

# Aerosol particle depolarization ratio at 1565 nm measured with a Halo Doppler lidar

Ville Vakkari[1,2], Holger Baars[3], Stephanie Bohlmann[4], Johannes Bühl[3], Mika Komppula[4], Rodanthi-Elisavet Mamouri[5,6], Ewan James O'Connor[1,7]

[1]Finnish Meteorological Institute, Helsinki, FI-00101, Finland
[2]Atmospheric Chemistry Research Group, Chemical Resource Beneficiation, North-West University, Potchefstroom, South Africa
[3]Leibniz Institute for Tropospheric Research, Leipzig, Germany
[4]Finnish Meteorological Institute, Kuopio, FI-70211, Finland
[5]Department of Civil Engineering and Geomatics, Cyprus University of Technology, Limassol, Cyprus
[6]ERATOSTHENES Centre of Excellence, Limassol, Cyprus
[7]Department of Meteorology, University of Reading, Reading, UK

*Correspondence to*: Ville Vakkari (ville.vakkari@fmi.fi)

**Abstract.** Depolarization ratio is a valuable parameter for lidar-based aerosol categorization. Usually, aerosol particle depolarization ratio is determined at relatively short wavelengths of 355 nm and/or 532 nm, but some multi-wavelength studies including longer wavelengths indicate strong spectral dependency. Here, we investigate the capabilities of Halo Photonics Stream Line Doppler lidars to retrieve the particle linear depolarization ratio at 1565 nm wavelength. We utilize collocated measurements with another lidar system, PollyXT at Limassol, Cyprus, and at Kuopio, Finland, to compare the
depolarization ratio observed by the two systems. For mineral dust-dominated cases we find typically a little lower depolarization ratio at 1565 nm than at 355 nm and 532 nm. However, for dust mixed with other aerosol we find higher depolarization ratio at 1565 nm. For polluted marine aerosol we find marginally lower depolarization ratio at 1565 nm compared to 355 nm and 532 nm. For mixed spruce and birch pollen we find a little higher depolarization ratio at 1565 nm compared to 532 nm. Overall, we conclude that Halo Doppler lidars can provide particle linear depolarization ratio at 1565
nm wavelength at least in the lowest 2-3 km above ground.

## 1 Introduction

Aerosols and their interactions with clouds remain the largest source of uncertainty in the Earth's radiative budget (IPCC, 2013). Remote sensing measurements with lidars enable continuous long-term observations of the vertical distribution of aerosol particles and clouds in the atmosphere, providing valuable information for improving our understanding of the global
climate system (e.g. Illingworth et al., 2015). Information on the vertical distribution of aerosols is highly important also for the aviation industry in case of hazardous aerosol emissions from e.g. volcanic eruptions (Hirtl et al., 2020).



Lidar measurements of aerosol optical properties at multiple wavelengths can be used to categorize elevated aerosol layers into different types such as mineral dust, smoke, marine aerosol or volcanic ash (e.g. Baars et al., 2017; Papagiannopoulos et al., 2018). One of the most important parameters for such aerosol typing is the depolarization ratio, which enables

distinguishing spherical and non-spherical particles from each other (e.g. Burton et al., 2012; Baars et al., 2017). Furthermore, the depolarization ratio can be used to quantify the contributions of different aerosol types to elevated layers (Mamouri and Ansmann, 2017). It is essential also for estimating vertical profiles of cloud condensation nuclei (CCN) and ice nucleating particle (INP) concentrations from remote sensing observations (Mamouri and Ansmann, 2016).

Currently, the particle linear depolarization ratio is most commonly measured at relatively short wavelengths of 355 nm

and/or 532 nm (e.g. Illingworth et al., 2015; Baars et al., 2016), though some lidar systems are capable of depolarization ratio measurement at longer wavelengths of 710 nm and 1064 nm (e.g. Freudenthaler et al., 2009; Burton et al., 2012). For instance, Burton et al. (2012) used the ratio of depolarization ratio at 1064 nm and 532 nm as part of their aerosol typing procedure. However, to our knowledge, aerosol particle depolarization ratio has not been reported at longer wavelengths than 1064 nm.

Previous studies on the spectral dependency of depolarization ratio between 355 nm and 1064 nm have shown a steep decrease in depolarization ratio from 532 nm to 1064 nm for elevated biomass burning aerosols (Burton et al., 2012, 2015; Haarig et al., 2018; Hu et al., 2019). On the contrary, mineral dust layers present increasing depolarization ratio with increasing wavelength (Gross et al., 2011; Burton et al., 2015) or a relatively weak maximum at 532 nm (Freudenthaler et al., 2009; Burton et al., 2015; Haarig et al., 2017). For some aerosol types, such as marine aerosol (Gross et al., 2011) and

volcanic ash (Gross et al., 2012), no spectral dependency was observed. However, volcanic ash mixed with boundary layer aerosol was observed with clearly lower depolarization ratio at 355 nm than at 532 nm (Gross et al., 2012).

Spectral dependency of the depolarization ratio has been attributed largely to the shape of the size distribution of polarizing aerosol particles. In smoke layers, the depolarization signal is probably due to non-spherical soot aggregates, which are in the size range of 100 nm to hundreds of nm and thus do not produce a large depolarization ratio at 1064 nm (Burton et al.,

2015; Haarig et al., 2018; Hu et al., 2019). On the other hand, mineral dust contains significant amounts of coarse mode particles (> 1 µm in diameter), which can explain the large depolarization ratio also observed at 1064 nm wavelength (Freudenthaler et al., 2009; Gross et al., 2011; Burton et al., 2015; Haarig et al., 2017). In aged dust layers, the faster removal of supermicron particles is thought to result in the depolarization ratio peaking at 532 nm (Freudenthaler et al., 2009; Gross et al., 2011; Burton et al., 2015; Haarig et al., 2017). In other words, spectral analysis of the depolarization ratio could permit

more in-depth diagnosis of coarse mode polarizing aerosol.

Halo Stream Line Doppler lidars are commercially available fibre-optic systems that operate at a wavelength of 1565 nm and can be equipped with a cross-polar receiver channel for measuring depolarization ratio (Pearson et al., 2009). Over the last few years these lidars have become widely used in wind and turbulence studies (e.g. Päschke et al., 2015; Vakkari et al., 2015; Tuononen et al., 2017; Manninen et al., 2018). Additionally, depolarization ratio measurements by Halo lidars have

been used to study cloud and precipitation phase (e.g. Achtert et al., 2015).



Now, recently developed post-processing (Vakkari et al., 2019) allows the utilization of significantly weaker signals from Halo Doppler lidars than previously. Therefore, the main aim of this paper is to assess the capabilities of Halo Doppler lidars in providing particle linear depolarization ratio measurements at 1565 nm wavelength. To do so, we utilize collocated Halo Doppler lidar and multiwavelength Raman lidar PollyXT observations during two measurement campaigns, where different polarizing aerosols were observed. Overall, the comparison indicates that Halo Doppler lidars can add another wavelength at 1565 nm to studies on the spectral dependency of particle linear depolarization ratio, at least in the lowest 2-3 km above ground.

## 2 Materials and Methods

Here we use data from two measurement campaigns where a Halo Photonics Doppler lidar and a PollyXT Raman lidar were collocated; at Kuopio, Finland, from 9 to 16 May 2016, and at Limassol, Cyprus, from 21 April to 22 May 2017. The campaigns represent quite different environments (Fig. 1) and enable the comparison of depolarization ratio at 1565 nm by the Halo instrument to depolarization ratio at 355 and 532 nm from PollyXT for a range of aerosol types. Furthermore, the campaigns were equipped with different devices of the Halo and PollyXT designs and thus potential differences between instrument individuals can be investigated.

The Vehmasmäki site (62.738°N, 27.543°E; 190 m a.s.l.) in Kuopio is a rural location surrounded by boreal forest (Bohlmann et al., 2019). The focus of the campaign in May 2016 was to investigate the capability to characterize the optical properties of airborne pollen with the multiwavelength Raman lidar PollyXT (Bohlmann et al., 2019). Here, we utilise one week of collocated measurements to compare Halo depolarization at 1565 nm to PollyXT during a spruce and birch pollination episode.

Limassol (34.675°N, 33.043°E; 22 m a.s.l.) is located at the southern shore of Cyprus in the Eastern Mediterranean. Measurements at Limassol were part of the Cyprus Clouds Aerosol and Rain Experiment (CyCARE; Ansmann et al., 2019) and were performed as a collaboration between Cyprus University of Technology (CUT), Limassol, and Leibniz Institute for Tropospheric Research (TROPOS), Leipzig. During April-May, several Saharan dust episodes were observed at Limassol in addition to the regional aerosol.

### 2.1 Halo Doppler lidar

Halo Photonics Stream Line lidars are commercially available 1565 nm pulsed Doppler lidars equipped with a heterodyne detector (Pearson et al., 2009). Halo Stream Line lidars emit linearly polarized light and the optical path is constructed with fibre-optic components, which can be equipped with a cross-polar receiver channel. The cross-polar channel is implemented through a fibre-optic switch between the normal receiver path and path with a fibre-optic polarizer. Thus, the measurement of the co- and cross-polar signals is not simultaneous, but consecutive in vertically-pointing mode. For instance, if the





integration time per ray is set to 7 s then co-polar signal is collected for 7 s and then cross-polar signal is collected during the next 7 s.

For research purposes, the most commonly used variants of Stream Line lidars are Stream Line, Stream Line Pro and Stream Line XR. The Stream Line and the more powerful Stream Line XR lidars enable full hemispheric scanning. The Streamline
Pro is designed without moving parts on the outside, which limits the scanning to a cone of 20° from vertical. All Stream Line variants can be used for depolarization ratio measurements, but an important difference between XR and other Stream Line versions is that the XR background noise level cannot be determined as accurately in the near range as for the non-XR versions (Vakkari et al., 2019). This difference is attributed to the more sensitive amplifier used in the Stream Line XR (Vakkari et al., 2019).

In this study we utilise vertically pointing measurements only from two Stream Line Pro systems. The operating specifications of these systems are given in Table 1. Stream Line lidars send and receive pulses through a single lens, which avoids issues with overlap and leads to a minimum range of 90 m due to impact of the outgoing pulse. At Vehmasmäki, we focused on boundary layer aerosol and set integration time per ray to 7 s and telescope focus to 2000 m. At Limassol, we expected to encounter elevated aerosol layers frequently and set integration time per ray to 11.5 s and telescope focus to
infinity. The integration time is set to balance between signal strength and good enough time resolution for retrievals of turbulent properties.

Halo Stream Line lidars provide three parameters along the beam direction: radial Doppler velocity, signal-to-noise ratio (SNR), and attenuated backscatter ($\beta$), which is calculated from SNR taking into account the telescope focus. A background check to determine background noise level is performed automatically once per hour. We post-processed SNR according to
Vakkari et al. (2019), which is essential to be able to further reduce the instrumental noise floor by averaging the SNR.

### 2.1.1 Halo depolarization ratio

We estimate the instrumental uncertainty in Halo Stream Line SNR from the standard deviation of SNR in the cloud- and aerosol-free part of the profile. Using mean values for the atmospheric number density taken from the U.S. Standard Atmosphere, 1976 (COESA 1976), the molecular backscatter coefficient at 1565 nm is about $1.9 \times 10^{-8}$ m$^{-1}$ sr$^{-1}$ at mean sea
level. Given the long wavelength and low pulse energy, no contribution from molecular scattering is observed in the signal, and the two-way atmospheric transmittance at 1565 nm is still 0.9994 at 2 km altitude above a lidar situated at mean sea level. Hence, the measured depolarization ratio can be safely assumed to represent the particle linear depolarization ratio.

In Fig. 2a, consecutive co- and cross-polar SNR profiles are presented, where aerosol signal is observed up to 800 m above ground level (a.g.l.) and a liquid cloud base is observed at 840 m a.g.l.. In liquid cloud the signal attenuates quickly and
above 1 km the profiles represent instrumental noise only. We use the measurements above 1 km to calculate standard deviations of co-polar SNR ($\sigma_{co}$) and cross-polar SNR ($\sigma_{cross}$). In Fig. 2b, raw depolarization ratio ($\delta$) is calculated simply as the ratio of cross-polar SNR to co-polar SNR and uncertainty is estimated from $\sigma_{co}$ and $\sigma_{cross}$ by Gaussian error propagation.





The construction of Halo Stream Line lidars does not enable user calibration of the depolarization ratio, unlike PollyXT lidars for instance (Engelmann et al., 2016). However, we can evaluate the Halo depolarization ratio at liquid cloud base.

Spherical cloud droplets do not polarize the directly back-scattered radiation and thus non-zero depolarization signal at liquid cloud base is an indication of incomplete extinction (or bleed-through) in the lidar internal polarizer. It should be noted, though, that multiple scattering results in increasing depolarization signal inside a liquid cloud (e.g. Liou and Schotland, 1971). This increase in in-cloud δ is clearly seen also in Fig. 2b. The magnitude of this effect depends on both cloud and lidar properties (e.g. Donovan et al., 2015); for Halo Stream Line lidars this effect is moderate as seen in Fig. 2b.

For the purpose of determining the polarizer bleed-through we minimize the effect of multiple scattering by considering δ only at the cloud base and determine average bleed-through from measurements in several clouds. Furthermore, we note that cloud-base δ should be determined from relatively high time resolution data to ensure that both co- and cross-polar measurements represent the same part of the cloud. Finally, it should be noted that, especially in higher latitudes, it is not always trivial to find purely liquid phase clouds. Typically, mixed-phase clouds can be distinguished in the histogram of

cloud-base δ as a secondary peak with higher δ than liquid clouds, but this requires the collection of data from a larger set of clouds.

To characterize the Halo polarizer bleed-through, we determined the depolarization ratio at liquid cloud base during both campaigns (Fig. 3). During the campaign at Limassol, we determined δ at cloud base on 25 April and on 2 May 2017. From the distribution in Fig. 3a, the bleed through is 0.011 ± 0.007 (mean ± standard deviation). At Vehmasmäki, we utilized

clouds on 13, 14 and 16 May 2016 as shown in Fig. 3b. At Vehmasmäki, the estimated bleed-through is 0.016 ± 0.009 (mean ± standard deviation).

We attribute the spread in the distributions in Fig. 3 mostly to variability of the clouds at the measurement sites and to the fact that co- and cross-polar measurements are consecutive and not simultaneous. Given that the cross-polar measurement channel is constructed with fibre-optic technology, we do not expect changes in the performance of the polarizer.

Considering the large natural variability of depolarization ratio (e.g. Illingworth et al., 2015; Baars et al., 2016) we find the spread of observations in Fig. 3 tolerable. The standard deviation in Fig. 3 is included in the uncertainty calculation of Halo depolarization ratio.

We account for the observed bleed-through (B) in calculating Halo depolarization ratio ($\delta_{1565}$) as

$$\delta = \frac{SNR_{cross} - B \cdot SNR_{co}}{SNR_{co}}, \tag{1}$$

where $SNR_{co}$ and $SNR_{cross}$ are observed co- and cross-polar SNR, respectively. Uncertainty in $SNR_{cross}$ corrected for bleed-through (i.e. numerator in Eq. 1) is estimated as

$$\sigma_{cross,B} = \sqrt{\sigma_{cross}^2 + (B \cdot SNR_{co})^2 \cdot \left(\frac{\sigma_B^2}{B^2} + \frac{\sigma_{co}^2}{SNR_{co}^2}\right)}, \tag{2}$$

where $\sigma_B$ is standard deviation of the distribution in Fig. 3. Finally, uncertainty in $\delta_{1565}$ taking into account instrumental noise and uncertainty in bleed-through is estimated as



$$\sigma_\delta = |\delta|\sqrt{\frac{\sigma_{cross,B}^2}{(SNR_{cross} - B\cdot SNR_{co})^2} + \frac{\sigma_{co}^2}{SNR_{co}^2}}.$$ (3)

## 2.2 PollyXT

PollyXT is a multiwavelength Raman lidar capable of depolarization ratio measurement at one or two wavelengths depending on instrument configuration (Baars et al., 2016; Engelmann et al., 2016). PollyXT emits simultaneously 355, 532 and 1064 nm wavelength pulses at a repetition frequency of 20 Hz. All PollyXT lidars are built with the same design, but 165 there are small differences in the number of receiver channels equipped in each individual system. A detailed description of PollyXT design is given by Baars et al. (2016) and Engelmann et al. (2016).

At Vehmasmäki, PollyXT was configured with elastic backscatter channels (355, 532 and 1064 nm), Raman-shifted channels at 387, 407 and 607 nm and a cross-polar channel at 532 nm (Bohlmann et al., 2019). Due to the biaxial construction of emission and detection units, complete overlap is reached at 800-900 m a.g.l. (Engelmann, et al., 2016) and 170 thus, only measurements > 800 m a.g.l. are utilized for this study (Bohlmann et al., 2019). The original spatial resolution is 30 m and temporal resolution 30 s for the Vehmasmäki system (Bohlmann et al., 2019).

At Limassol, PollyXT operated the same receiver channels as the Vehmasmäki system had and additionally a cross-polar channel at 355 nm, together with a near-range telescope with 355 and 532 nm receiver channels. The near-range channels enable retrieval of optical properties down to 150 m a.g.l. (Engelmann et al., 2016). Raw spatial resolution is 7.5 m and 175 temporal resolution, 30 s.

During night-time, the Raman-method (Ansmann et al., 1992) is used to retrieve aerosol optical properties from the raw signals. For daytime measurements, the method of Klett (1981) can be utilised. Here, we present only measurements when the Raman-method was applied. The calibration of depolarization ratio was performed at both Vehmasmäki and Limassol using the so-called Δ90°-method (Freudenthaler, 2016) and the relative uncertainty in particle linear depolarization ratio was 180 estimated to be 10 %.

## 2.3 Auxiliary data

Air mass history was estimated with the Hybrid Single-Particle Lagrangian Integrated Trajectory model HYSPLIT (Stein et al., 2015). HYSPLIT was run through the READY website (Rolph et al., 2017) using the NCEP Global Data Assimilation System (GDAS) meteorology at 0.5° horizontal resolution. 96 h back-trajectories were calculated arriving at the elevation of 185 aerosol layers of interest.

## 3 Results

In this Section we analyze observations of dust, marine and pollen aerosols during the Limassol and Vehmasmäki campaigns, where said aerosol types were observed simultaneously with Halo and PollyXT lidars. Dust and marine aerosols





were observed during the Limassol campaign in Eastern Mediterranean and pollen was observed during the Vehmasmäki
campaign in a boreal forest region in Finland. We conclude this section with an overall comparison of depolarization ratio
measurements with the two instruments.

### 3.1 Elevated dust layers

#### 3.1.1 Limassol 21 April 2017

Right at the beginning of Halo measurements at Limassol on 21 April 2017, several elevated layers were observed as seen in
Fig. 4. Although Halo can observe elevated layers up to 6 km a.g.l. on this day, the signal is too weak to retrieve their
depolarization ratio. This is clearly visible in the uncertainty in the Halo depolarization ratio in Fig. 4c. At 300 s integration
time (i.e. 10 minutes of alternating co- and cross-polar measurement), the depolarization ratio can be determined up 1-1.5 km
a.g.l. with $\sigma_\delta < 0.05$ on this day (Fig. 4d). The depolarization ratio can be retrieved also for the relatively strong elevated
layer at 3 km a.g.l. during the morning hours (Fig. 4d).

Increasing both temporal and spatial averaging enables the utilization of some of the weaker signals. Fig. 5 presents profiles
of the Halo and PollyXT depolarization ratio, where both are averaged over 1.5 h (20:00 – 21:30 UTC) and smoothed
vertically with a 300 m running mean. In the lowest layer < 1 km a.g.l., practically no difference is observed in the
depolarization ratio at the different wavelengths. Back-trajectory calculations (Fig. 6) indicate this layer to be mostly
regional air from Eastern Mediterranean and the relatively large lidar ratio is in the range of observations of smoke or smoke
mixed with dust (e.g. Gross et al., 2011; Baars et al., 2016). On the other hand, for the layer from 1.5 km to 2 km a.g.l. a
clear increase in δ with increasing wavelength is observed. For this layer air mass history indicates origins over Northern
Africa (Fig. 6) and the lidar ratio (42±4 at 355 nm, 47±5 at 532 nm) is in the range of dust (Ansmann et al., 2011). For this
layer the mean (± standard deviation) δ at 355 nm, 532 nm and 1565 nm are 0.19±0.008, 0.23±0.008 and 0.29±0.008,
respectively. Above 2 km a.g.l., the uncertainty in δ at 1565 nm increases rapidly and is not used for quantitative analysis
here.

#### 3.1.2 Limassol 27 April 2017

Stronger elevated aerosol layers were observed at Limassol on 27 April 2017. On this day, depolarization ratio can be
retrieved by Halo up to 3 km a.g.l. (Fig. 7). For an averaging period of 01:25-02:30 UTC, depolarization ratio is retrieved for
the elevated layer at 1600-2200 m a.g.l.. For this layer, the depolarization ratio at 1565 nm is 0.30±0.005, which is a little
lower than for the shorter wavelengths: 0.36±0.01 at 355 nm and 0.34±0.002 at 532 nm, respectively. For this layer, the air
mass history indicates southerly origins.
On the same day (27 April 2017) at 19:00-20:00 UTC, the depolarization ratio can be retrieved from the surface up to 2.6 km
a.g.l. (Fig. 8). in the lowest 500 m, depolarization ratio at 1565 nm is clearly higher than at the shorter wavelengths,
suggesting a mixture of larger mineral dust particles with smaller particles of lower depolarization ratio. For the layer at



1500-2500 m a.g.l., practically no wavelength-dependency is observed for depolarization ratio, indicating that backscatter at all wavelengths is dominated by the same aerosol. The layer-averaged depolarization ratios are 0.31±0.006, 0.33±0.005 and 0.32±0.008 at 355 nm, 532 nm and 1565 nm, respectively. This high depolarization ratio and lidar ratio of 47±5 at 355 nm (38±3 at 532 nm) indicate almost pure dust (Ansmann et al., 2011; Baars et al., 2016). Air mass history, on the other hand, indicates northerly or north-westerly origins at both 2 km a.g.l. and at the surface (Fig. 9).

### 3.2 Polluted marine aerosol

On 20 May 2017 at Limassol, very low aerosol depolarization ratio is observed throughout the day as seen in Fig. 10. During the morning and afternoon liquid clouds are observed but during the evening Raman retrievals with PollyXT were possible. Fig. 11 presents Halo depolarization ratio profiles averaged for the duration of the PollyXT retrieval at 19:54-21:30 UTC. For the surface layer (up to 1 km a.g.l.), a small decrease in depolarization ratio with increasing wavelength is observed. The layer-averaged depolarization ratios are 0.03±0.01, 0.015±0.002 and 0.009±0.003 at 355 nm, 532 nm and 1565 nm, respectively. The layer-averaged lidar ratio at 355 nm is 39±4 sr, whereas the lidar ratio at 532 nm is very noisy at 47±35 sr. The low depolarization ratio is typical of marine aerosol, smoke and pollution (Gross et al., 2011; Illingworth et al., 2015). The 355 nm lidar ratio lies between the values reported for marine aerosol and smoke (Illingworth et al., 2015).

Above 1 km a.g.l., an optically thin aerosol layer is observed (Fig. 11). Halo indicates a higher depolarization ratio for this layer than at the surface, but the signal is so weak that the uncertainty in depolarization ratio at 1565 nm becomes very large (Fig. 11b). Back-trajectories arriving over Limassol at 21 UTC indicate different, but mostly northerly origins for the air mass at 500 m a.g.l. and at 2 km a.g.l. (Fig. 12).

### 3.3 Pollen in boreal forest

On 15 May 2016, substantial amounts of spruce and birch pollen were observed at Vehmasmäki with both an in-situ sampler and the PollyXT lidar (Bohlmann et al., 2019). The presence of more polarizing spruce pollen (Bohlmann et al., 2019) in the boundary layer is observed also with Halo lidar as seen in Fig. 13d. However, the backscatter (Fig. 14a) is low compared to the case studies presented for Limassol and the low signal results in significant noise in the lidar ratio (Fig. 14c).

Comparing the depolarization ratios measured with Halo and PollyXT (Fig. 14b) shows a nearly constant depolarization ratio at 1565 nm, while the depolarization ratio at 532 nm decreases with height. At 1565 nm, the Halo signal is probably dominated by pollen grains, which are tens of micrometres in diameter. At 355 nm and 532 nm wavelengths, the backscatter is increasing with height (Fig. 14a) and thus the decreasing depolarization ratio at 532 nm may reflect an increasing fraction of signal from non-pollen aerosol with increasing height. For the layer from 800 m to 1 km a.g.l. in Fig. 14, the mean depolarization ratios are 0.236±0.009 and 0.269±0.005 at 532 nm and 1565 nm, respectively.





### 3.4 Overview of depolarization ratio wavelength dependency

An overall comparison of the depolarization ratio at different wavelengths for the Limassol and Vehmasmäki campaigns is presented in Fig. 15, where the Halo vertical resolution of 30 m has been smoothed with a 300 m running mean. The original time resolution observations by Halo have been averaged to match the temporal resolution of PollyXT Raman retrievals (ranging from 45 min to 2 h).

In Fig. 15a, three regions can be observed in the scatterplot. For $\delta_{532} < 0.05$, $\delta_{1565}$ matches very closely with the shorter
wavelength. For $\delta_{532}$ ranging from 0.05 to 0.25, $\delta_{1565}$ is systematically larger than $\delta_{532}$. For $\delta_{532} > 0.3$, $\delta_{1565}$ is lower than the depolarization ratio at the shorter wavelength. A very similar pattern is present in Fig. 15b: for $\delta_{355} < 0.05$, $\delta_{1565}$ matches $\delta_{355}$ closely; for $\delta_{355}$ ranging from 0.05 to 0.25, $\delta_{1565}$ is larger than $\delta_{355}$ and for $\delta_{355} > 0.3$, $\delta_{1565}$ is lower than $\delta_{355}$. Even comparing the two shorter wavelengths (Fig. 15c), similar regions appear: for $\delta_{355} < 0.05$, $\delta_{532}$ is lower than $\delta_{355}$; for $\delta_{355}$ ranging from 0.1 to 0.3 depolarization ratio is on average equal on both wavelengths and for $\delta_{355} > 0.3$, $\delta_{532}$ is lower than $\delta_{355}$.

Figs. 15a-c show also similar correlations between the depolarization ratios at different wavelengths. Therefore, bearing in mind the similar patterns in all three scatterplots in Figs. 15a-c, we consider the scatter to originate mainly from the atmospheric aerosol properties rather than in instrumental effects. For instance, any bias in the estimated bleed-through in the Halo polarizer would show up as bias in Fig. 15a and 15b. However, such bias is not present in the cases when $\delta_{355}$ and/or $\delta_{532}$ are low.

Considering the sources at Limassol during the campaign, the higher $\delta_{1565}$ for intermediate depolarization ratios ranging from 0.1 to 0.25 likely represents mixtures of dust with other aerosol types. A mixture of coarse, polarizing dust with less polarizing and smaller aerosol would result in the observed spectral dependency of depolarization ratio. For aged dust-dominated cases, lower depolarization ratios at longer wavelength could be due to the faster removal of coarse particles compared to submicron aerosol (e.g. Burton et al., 2015). In any case, the observed wavelength dependency in Figs. 15a-c
for large $\delta$ suggests that, for dust-dominated cases, smaller particle sizes have, on average, higher depolarization ratio at Limassol.

Another type of polarizing aerosol, i.e. pollen, was observed with a collocated Halo and PollyXT at Vehmasmäki (Bohlmann et al., 2019). Comparatively low signal levels, together with 800 m minimum range for the PollyXT system at Vehmasmäki (Bohlmann et al., 2019), reduce the amount of data available for comparison of Halo and PollyXT depolarization ratio
during the campaign (Fig. 15d). During this campaign, the depolarization ratio at 1565 nm is a little larger than at 532 nm, but the difference is small compared to the scatter observed at Limassol.

A further look into the distribution and spectral dependency of the depolarization ratio at Limassol is presented in Fig. 16. In Figs. 16a and 16b, the 2D-histograms of depolarization ratio show that both 532 nm and 1565 nm wavelengths present a bi-modal distribution below 1 km a.g.l.. In other words, there are also less polarizing aerosols frequently present in the lowest 1
km in addition to dust and dusty mixtures with depolarization ratio > 0.2. However, above about 1.5-2 km a.g.l., almost all





retrievals indicate dust or dusty mixtures. Note that the vertical extent of the data is limited by the sensitivity of the Halo instrument, as Figs. 16a and 16b are limited to cases when both wavelengths are available.

In Figs. 16c and 16d, the ratio of depolarization ratios at 1565 nm and 532 nm exhibits clear height-dependency. Above about 1.5 km a.g.l., the majority of the observations present a lower depolarization ratio at 1565 nm than at 532 nm, while
below 1.5 km a.g.l., the depolarization ratio is higher at the longer wavelength. In previous studies (Freudenthaler et al., 2009; Gross et al., 2011; Burton et al., 2015; Haarig et al., 2017), a lower depolarization ratio at longer wavelengths has been attributed to faster removal of coarse mode dust. However, our observations indicate the presence of a small coarse mode, probably mineral dust, for sub-1.5 km aerosols most of the time at Limassol.

## 4 Discussion

The majority of aerosol depolarization ratio measurements have been carried out at relatively short wavelengths (355 nm and 532 nm) with only a few previous studies investigating the spectral dependency including 710 nm (Freudenthaler et al., 2009; Gross et al., 2011) and/or 1064 nm (Freudenthaler et al., 2009; Burton et al., 2012, 2015; Haarig et al., 2017, 2018; Hu et al., 2019). In this study we have for the first time determined aerosol particle depolarization ratios at a wavelength of 1565 nm.

From an instrumental point of view, the Halo Doppler lidar depolarization ratio seems to be of comparable quality to PollyXT depolarization ratio when the aerosol signal is strong. However, Halo has a much less powerful laser than PollyXT, which limits significantly the range of usable signal. On the other hand, Halo Doppler lidars are capable of independent operation for months and are therefore suitable for operational use in meteorological measurement networks.

The integration time and range gate length are adjustable in Halo firmware and prolonging these parameters would increase
the sensitivity of the system. However, high spatial and temporal resolution are preferable for utilizing the Doppler capabilities of Halo lidars. Inspecting the internal polarizer performance at liquid cloud base also requires a higher resolution. Overall, the configuration of a Halo Doppler lidar needs to be considered individually for the aims of each measurement campaign.

The spectral dependency that we observed for 355 nm, 532 nm and 1565 nm particle linear depolarization ratio agrees
reasonably well with previous spectral analyses for similar aerosol types as seen in Table 2. For mineral dust depolarization ratio, both decreasing and increasing trends with increasing wavelength have been observed previously (Table 2). This is the case for our observations at Limassol as well, though on average $\delta_{1565}$ tends to be a little lower than $\delta_{532}$ (Fig. 16). Probably, the spectral dependency of mineral dust depolarization ratio depends on both the age of the dust and the origin of the dust.

Wavelength-dependent changes in mineral dust depolarization ratio are small compared to elevated smoke layers, which can
help to distinguish between these two aerosol types (Burton et al., 2012). For elevated smoke, a strong decrease in depolarization ratio has been reported from > 0.20 at short wavelengths to $\delta < 0.05$ at 1064 nm (Burton et al., 2015; Haarig et



al., 2018; Hu et al., 2019). Thus, adding a depolarization ratio measurement at 1565 nm can provide added value to the commonly-used measurements at 355 nm and 532 nm wavelengths.

For marine aerosols, the depolarization ratio is small and has practically no spectral dependency (Gross et al., 2011), which

is what we observed at Limassol. For the mixture of spruce and birch pollen at Vehmasmäki, the differences in depolarization ratio at 532 nm and 1565 nm are small.

## 5 Conclusions

In this paper we report for the first time remote sensing measurements of atmospheric aerosol particle linear depolarization ratio at a wavelength of 1565 nm. Using observations at liquid cloud base we have been able to characterize the Halo

Doppler lidar polarizer bleed-through with sufficient accuracy to obtain useful depolarization ratio measurements; uncertainty in the bleed-through is propagated to the depolarization ratio measurement. A comparison of two different Halo Doppler lidar systems with two PollyXT systems during collocated measurements at Limassol, Cyprus, and Kuopio - Vehmasmäki, Finland, show good agreement between the lidar systems. The agreement between the instruments is remarkably good considering the large wavelength difference: the PollyXT depolarization ratio is retrieved at 355 nm and/or

532 nm. However, given the much lower laser energy in Halo Doppler lidars, it is not surprising that the vertical extent of usable depolarization ratio is much lower than for PollyXT.

For relatively fresh mineral dust, we find particle linear depolarization ratios at 1565 nm ranging from 0.29 to 0.32, which is in good agreement with previous observations, including measurements at 710 nm and 1064 nm wavelengths (Freudenthaler et al., 2009; Gross et al., 2011; Burton et al., 2015; Haarig et al., 2017). For polluted marine aerosol we observed very low

depolarization ratio of 0.009 at 1565 nm with a small decrease with increasing wavelength. Spruce and birch pollen depolarization ratio has been characterized only recently at 532 nm (Bohlmann et al., 2019). Our measurements indicate a slightly higher depolarization ratio of 0.27 at 1565 nm compared to 0.24 at 532 nm. Overall, our results indicate that Halo Doppler lidars can add another wavelength at 1565 nm to studies on the spectral dependency of particle linear depolarization ratio, at least in the lowest 2-3 km above ground.

## 335 Data availability

Processed lidar data are available upon request from the authors. Level 0 PollyXT observations are available at http://polly.rsd.tropos.de/ (last access 18 August 2020). Trajectory model HYSPLIT and GDAS meteorological data are available at https://www.ready.noaa.gov/HYSPLIT.php (last access 18 August 2020).



**Author contribution**

Conceptualization and formal analysis, V.V.; investigation, H.B, S.B., J.B., M.K., R.M., E.O.C.; data curation, V.V., H.B, S.B., M.K., E.O.C.; writing—original draft preparation, V.V.; writing—review and editing, V.V.

**Competing interests**

The authors declare that they have no conflict of interest.

**Acknowledgments**

This study was funded by The National Emergency Supply Agency of Finland. The Vehmasmäki pollen study and data evaluation were supported by the Academy of Finland (project no. 310312). The Limassol, Cyprus observations have been supported by the SIROCCO project (grant no. EXCELLENCE/1216/0217) co-funded by the Republic of Cyprus and the structural funds of the European Union for Cyprus through the Research and Innovation Foundation and EXCELSIOR project that received funding from the European Union [H2020-WIDESPREAD-04-2017:Teaming Phase2] project under
grant agreement no. 857510, and from the Republic of Cyprus. The authors gratefully acknowledge the NOAA Air Resources Laboratory (ARL) for the provision of the HYSPLIT transport and dispersion model and READY website (https://www.ready.noaa.gov) used in this publication.

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



**Table 1. Specifications of Halo Doppler lidars used in this study.**

| | |
|---|---|
| Wavelength | 1565 nm |
| Pulse repetition rate | 15 kHz |
| Pulse energy | 20 μJ |
| Pulse duration | 0.2 μs |
| Nyquist velocity | 20 m s$^{-1}$ |
| Sampling frequency | 50 MHz |
| Velocity resolution | 0.038 m s$^{-1}$ |
| Points per range gate | 10 |
| Range resolution | 30 m |
| Maximum range | 9600 m |
| Lens diameter | 8 cm |
| Lens divergence | 33 μrad |
| Telescope | monostatic optic-fibre coupled |

**Table 2. Spectral dependency of depolarization ratio for dust, marine aerosol and pollen.**

| | | Depolarization ratio | | | | |
|---|---|---|---|---|---|---|
| | Time and origin | 355 nm | 532 nm | 710 nm | 1064 nm | 1565 nm |
| **This study,** Limassol | 21 April 2017 20:00-21:30; Saharan dust | 0.19±0.008 | 0.23±0.008 | | | 0.29±0.008 |
| | 27 April 2017 01:25-02:33; dust (Egypt) | 0.36±0.01 | 0.34±0.002 | | | 0.30±0.005 |
| | 27 April 2017 19:00-20:00; dust (Turkey) | 0.31±0.006 | 0.33±0.005 | | | 0.32±0.008 |
| Haarig et al. (2017) | Barbados 2013, 2014; Saharan dust | 0.252±0.030 | 0.280±0.020 | | 0.225±0.022 | |
| Burton et al. (2015) | US 2014; Saharan dust | 0.209±0.015 | 0.304±0.005 | | 0.270±0.005 | |
| | Mexico Chihuahua 2013; local dust | 0.225±0.041 | 0.373±0.014 | | 0.383±0.006 | |
| Gross et al. (2011) | Cape Verde 2008; Saharan dust | 0.24 – 0.27 | 0.29 – 0.31 | 0.36 – 0.40 | | |
| Freudenthaler | Morocco 2006; | 0.24 – 0.28 | 0.31±0.03 | 0.26 – 0.30 | 0.27± 0.04 | |



| | | | | |
|---|---|---|---|---|
| et al. (2009) | Saharan dust | | | |
| **This study,** Limassol | 20 May 2017 19:55-21:30; polluted marine | 0.03±0.01 | 0.015±0.002 | 0.009±0.003 |
| Gross et al. (2011) | Cape Verde 2008; marine | 0.02±0.01 | 0.02±0.02 | |
| **This study,** Vehmasmäki | 15 May 2016 19:00-21:00; spruce and birch pollen | | 0.236±0.009 | 0.269±0.005 |

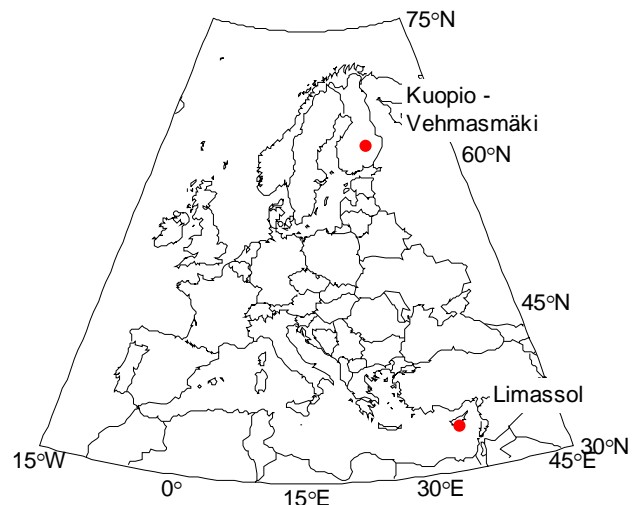

**Figure 1: Locations of Vehmasmäki and Limassol measurement sites.**




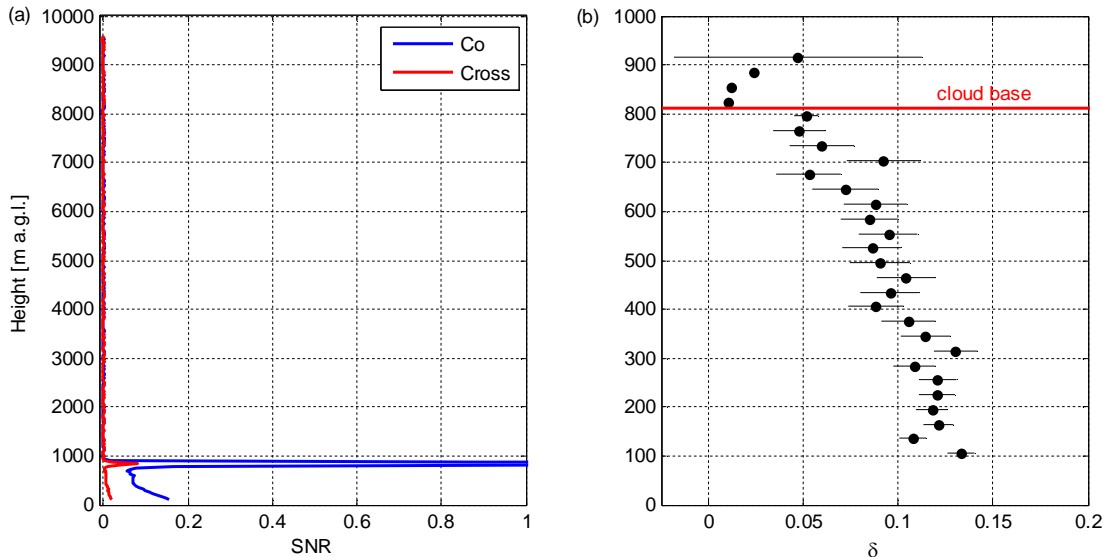

**Figure 2: Profiles at Limassol, Cyprus, on 2 May 2017 at 12:08 UTC. (a) Co- and cross-polar SNR. A liquid cloud at approx. 800 m a.g.l. results in full attenuation of signal. Below cloud layer aerosol signal is visible. Above 1 km variability in SNR is due to instrumental noise only. (b) Depolarization ratio profile up to 1 km a.g.l. calculated from profiles in panel (a). Error bars represent**
**uncertainty due to instrumental noise estimated from SNR at > 1 km a.g.l. in panel (a).**

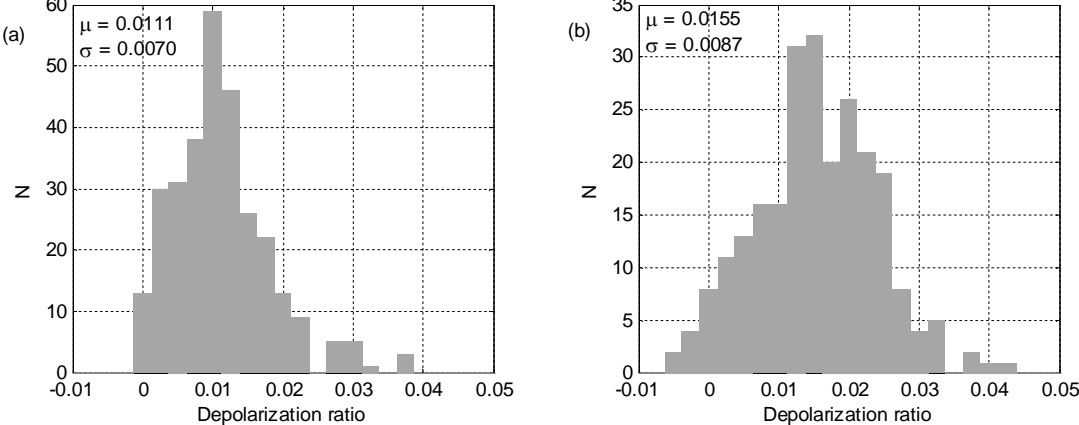

**Figure 3: Depolarization ratio at liquid cloud base measured with Halo Doppler lidar. (a) Distribution of cloud base depolarization ratio at Limassol. (b) Distribution of cloud base depolarization ratio at Vehmasmäki.**



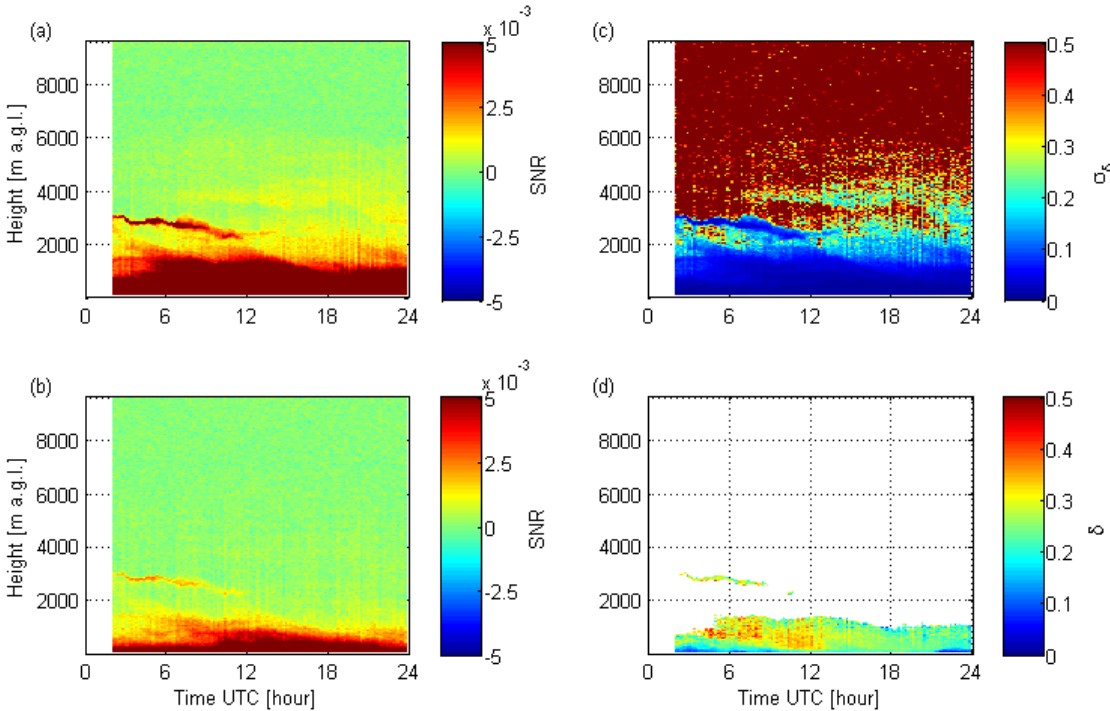


**Figure 4: Limassol 21 April 2017 measurements with Halo Doppler lidar. (a) Time series of co-polar SNR at 300s integration time. (b) Time series of cross-polar SNR at 300s integration time. (c) Uncertainty in depolarization ratio. (d) Depolarization ratio filtered with σ_δ < 0.05.**





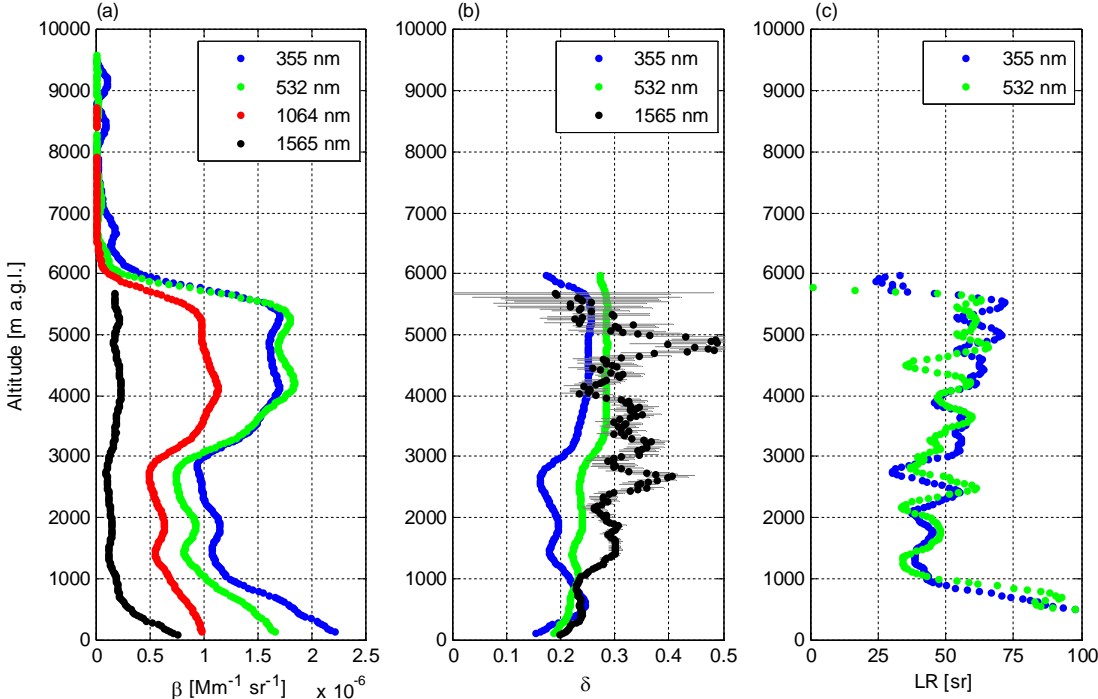

**Figure 5: Averaged profiles at Limassol on 21 April 2017 20:00-21:30 UTC. All profiles have been smoothed by 300 m running mean. (a) Backscatter by PollyXT (wavelengths 355 – 1064 nm) and attenuated backscatter by Halo (1565 nm). (b) Depolarization ratio. Error bars represent measurement uncertainty. (c) Lidar ratio. For PollyXT $\beta_{355}$, $\beta_{532}$, $LR_{355}$ and $LR_{532}$ near range telescope is used for data < 900 m a.g.l..**

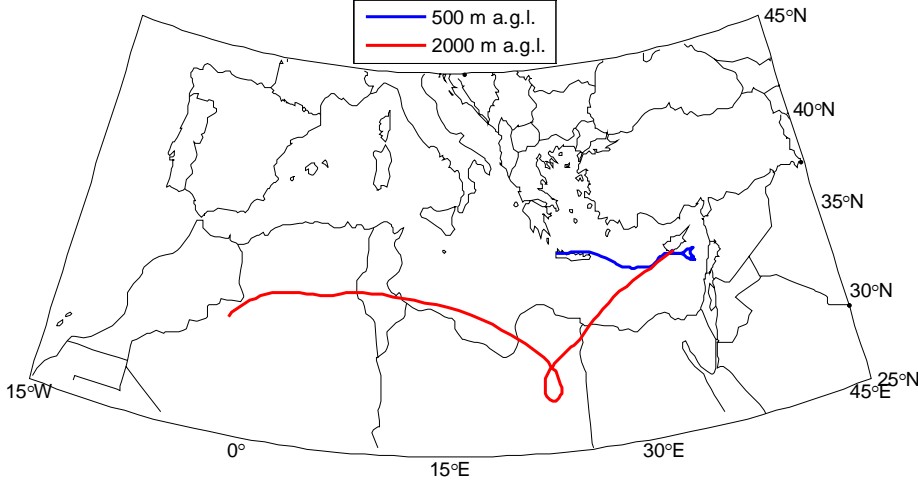

**Figure 6: 96-hour back-trajectories arriving at Limassol on 21 April 2017 at 21:00 UTC.**





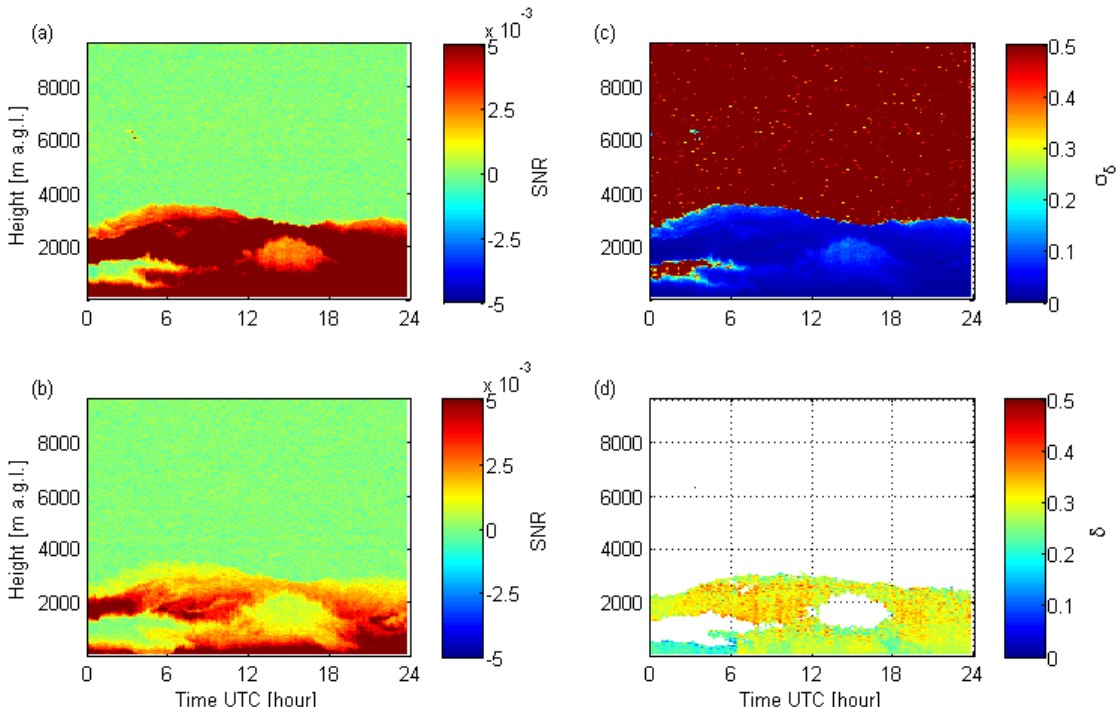

**Figure 7: Limassol 27 April 2017 measurements with Halo Doppler lidar. (a) Time series of co-polar SNR at 300s integration time. (b) Time series of cross-polar SNR at 300s integration time. (c) Uncertainty in depolarization ratio. (d) Depolarization ratio filtered with $\sigma_\delta < 0.05$.**





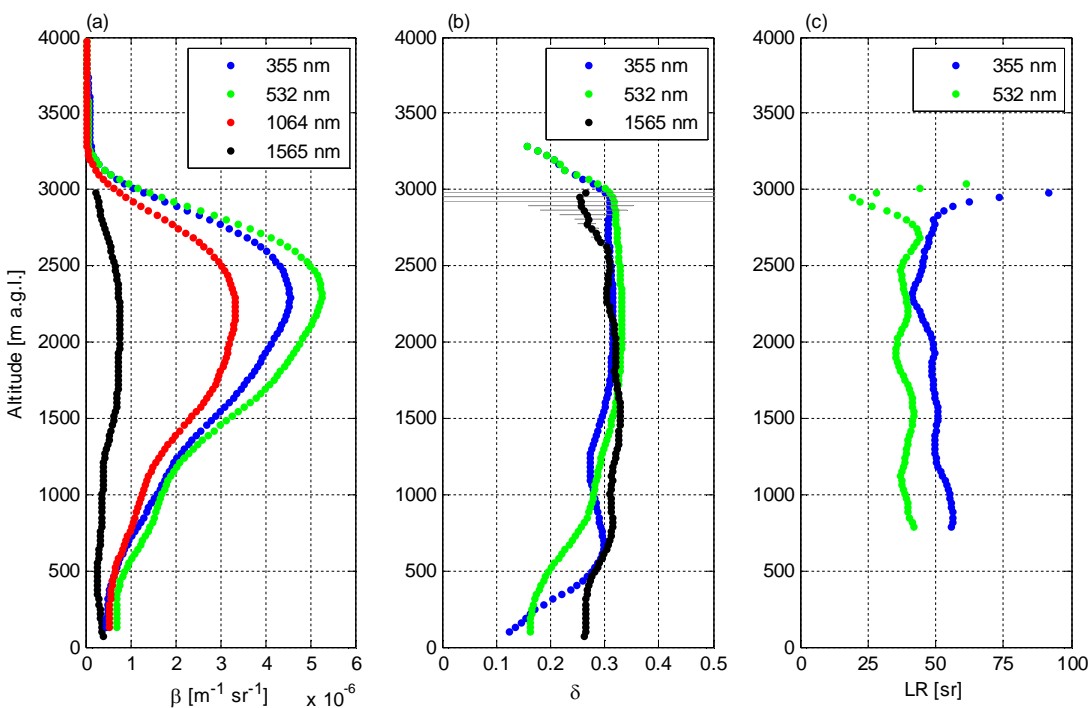

**Figure 8:** Averaged profiles at Limassol on 27 April 2017 19:00-20:00 UTC. All profiles have been smoothed by 300 m running mean. (a) Backscatter by PollyXT (wavelengths 355 – 1064 nm) and attenuated backscatter by Halo (1565 nm). (b) Depolarization ratio. Error bars represent measurement uncertainty. (c) Lidar ratio. For PollyXT $\beta_{355}$ and $\beta_{532}$ near range telescope is used for data < 900 m a.g.l..

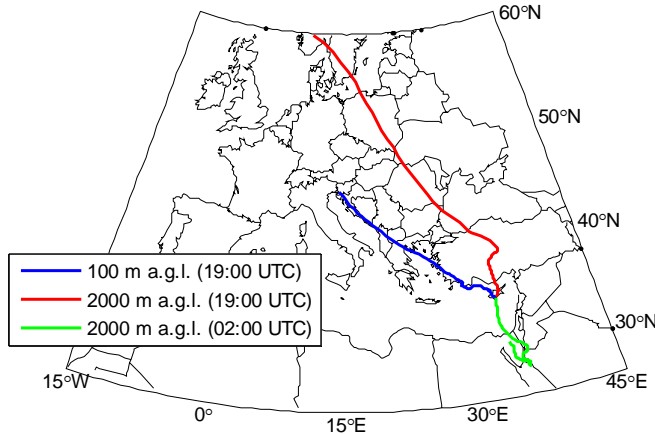

**Figure 9:** 96-hour back-trajectories arriving at Limassol on 27 April. Back-trajectories arriving at 19:00 and 02:00 UTC are included.



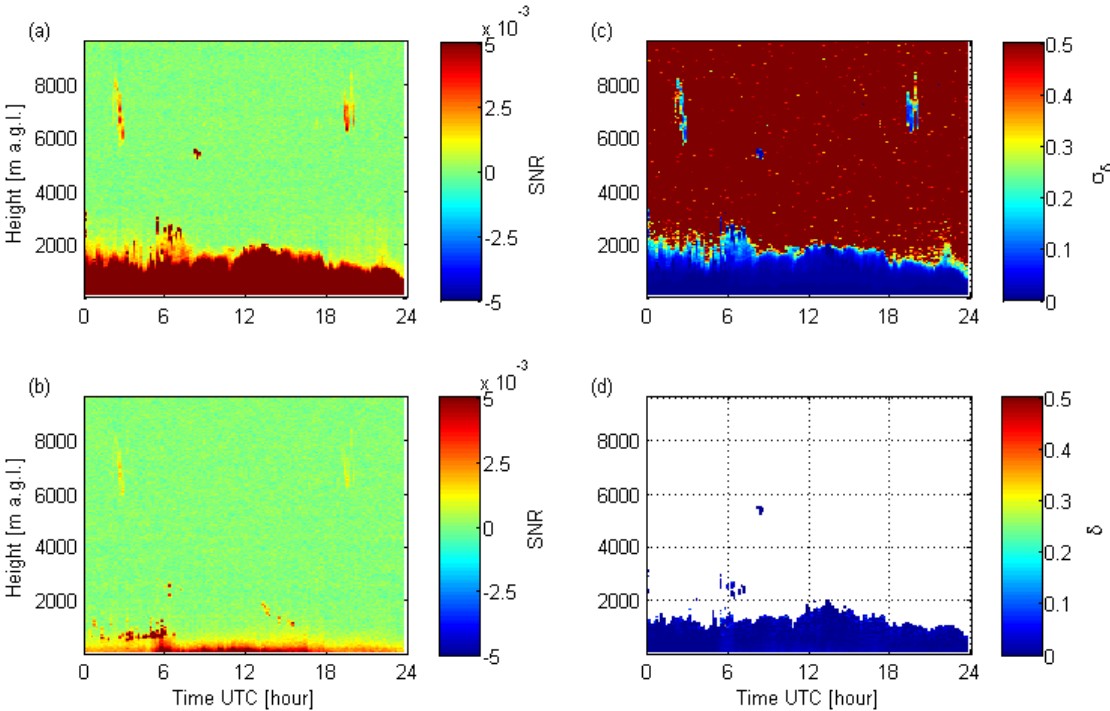

**Figure 10: Limassol 20 May 2017 measurements with Halo Doppler lidar. (a) Time series of co-polar SNR at 300s integration time.**
**(b) Time series of cross-polar SNR at 300s integration time. (c) Uncertainty in depolarization ratio. (d) Depolarization ratio**
**filtered with $\sigma_\delta < 0.05$.**



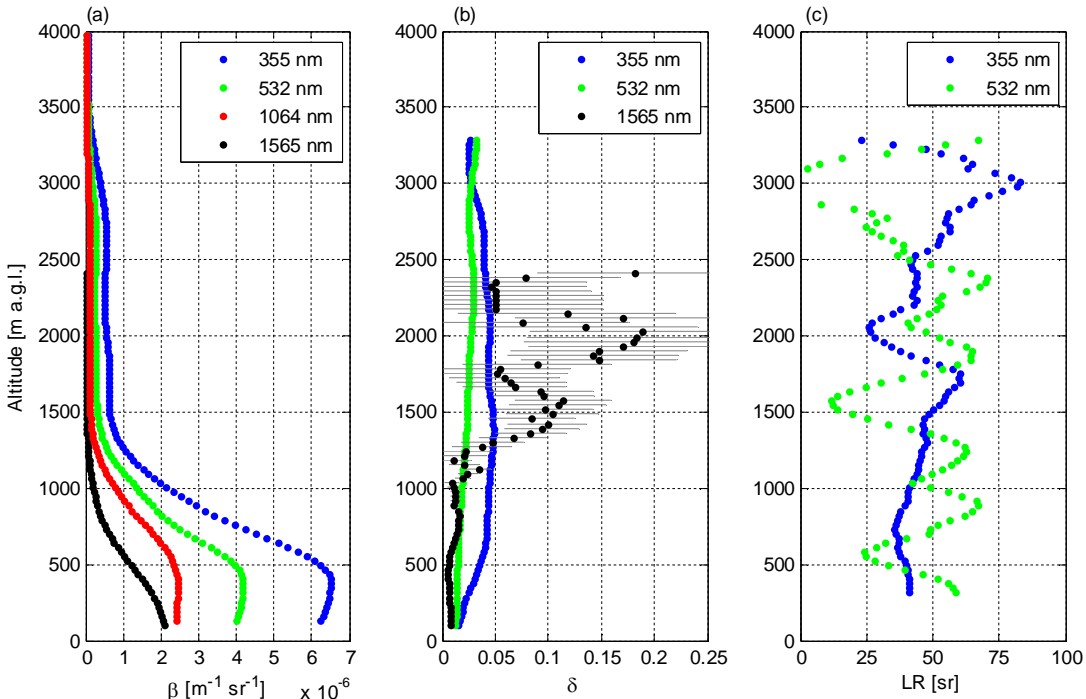

**Figure 11: Averaged profiles at Limassol on 20 May 2017 19:55-21:30 UTC. All profiles have been smoothed by 300 m running mean. (a) Backscatter by PollyXT (wavelengths 355 – 1064 nm) and attenuated backscatter by Halo (1565 nm). (b) Depolarization ratio. Error bars represent measurement uncertainty. (c) Lidar ratio. For PollyXT $\beta_{355}$, $\beta_{532}$, $LR_{355}$ and $LR_{532}$ near range telescope is used for data < 900 m a.g.l..**

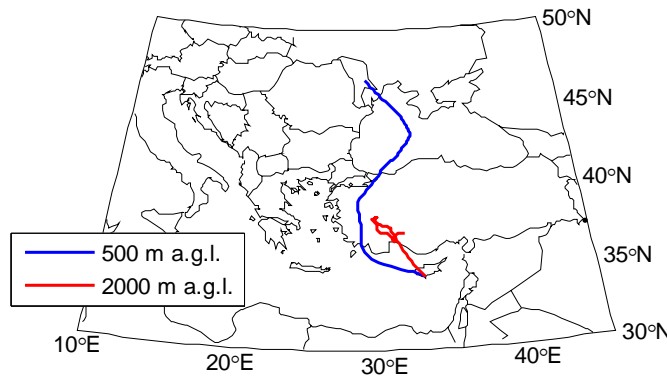

**Figure 12: 96-hour back-trajectories arriving at Limassol on 20 May 2017 at 21:00 UTC.**




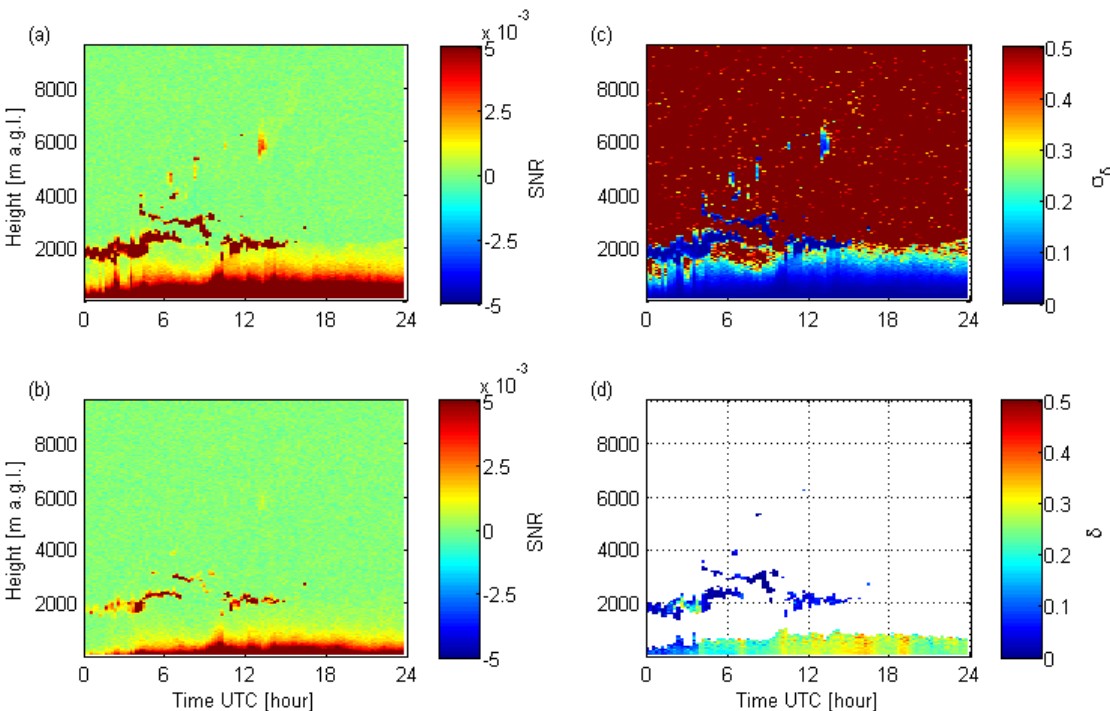

**Figure 13: Vehmasmäki 15 May 2016 measurements with Halo Doppler lidar. (a) Time series of co-polar SNR at 350 s integration time. (b) Time series of cross-polar SNR at 350 s integration time. (c) Uncertainty in depolarization ratio. (d) Depolarization ratio filtered with $\sigma_\delta < 0.05$.**





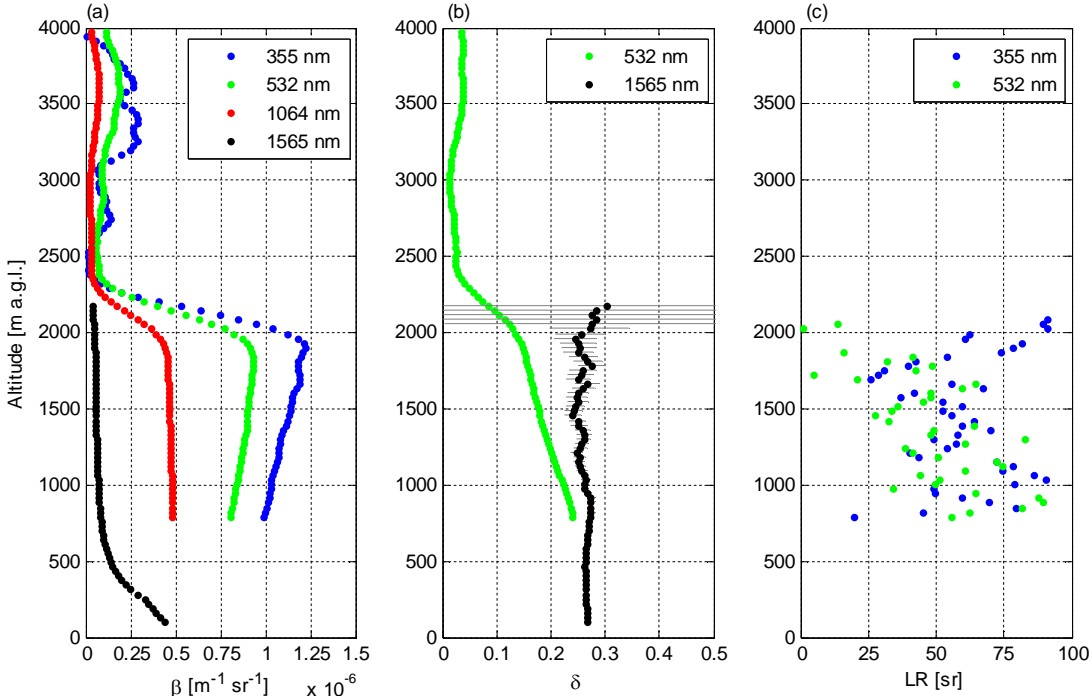

**Figure 14: Averaged profiles at Vehmasmäki on 15 May 2016 19:00-21:00 UTC. All profiles have been smoothed by 300 m running**
**mean. (a) Backscatter by PollyXT (wavelengths 355 – 1064 nm) and attenuated backscatter by Halo (1565 nm). (b) Depolarization ratio. Error bars represent measurement uncertainty. (c) Lidar ratio.**





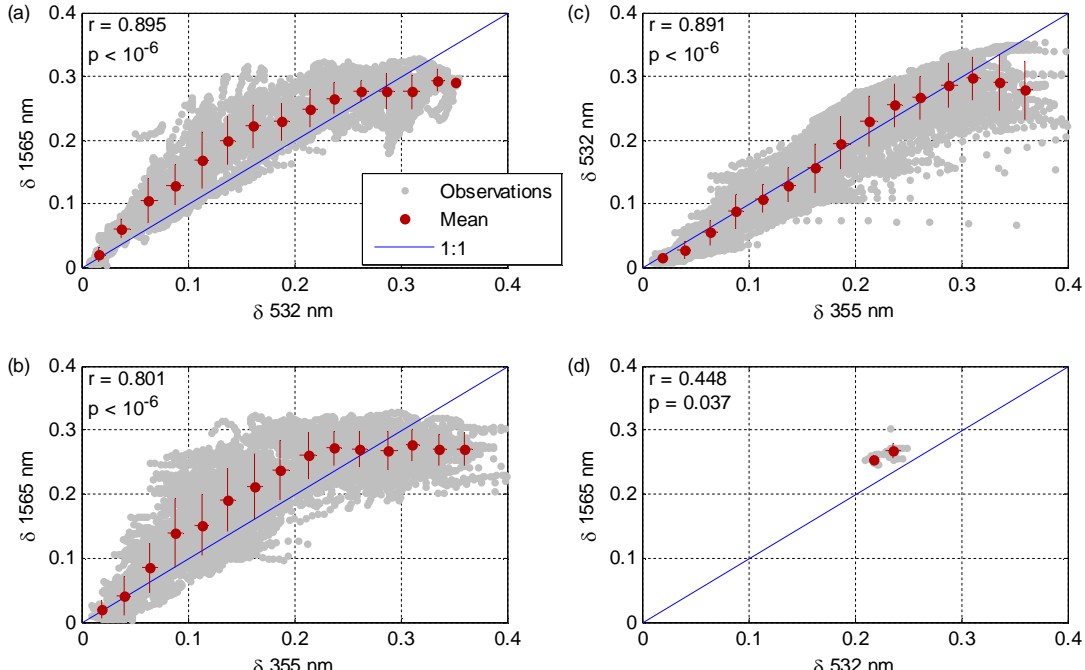

**Figure 15: Comparison of depolarization ratio at different wavelengths. Observations represent 30 m vertical resolution and have**
**been smoothed by 300 m running mean. Only data for $\sigma_\delta < 0.01$ (at 1565 nm wavelength) is included. Mean is calculated at intervals of 0.025 on the x-axis and errorbars indicate standard deviation. (a) Depolarization ratio at 1565 nm (Halo) vs. depolarization ratio at 532 nm (PollyXT) at Limassol. (b) Depolarization ratio at 1565 nm vs. depolarization ratio at 355 nm (PollyXT) at Limassol. (c) Depolarization ratio at 532 nm vs. depolarization ratio at 355 nm at Limassol. (d) Depolarization ratio at 1565 nm vs. depolarization ratio at 532 nm at Vehmasmäki.**



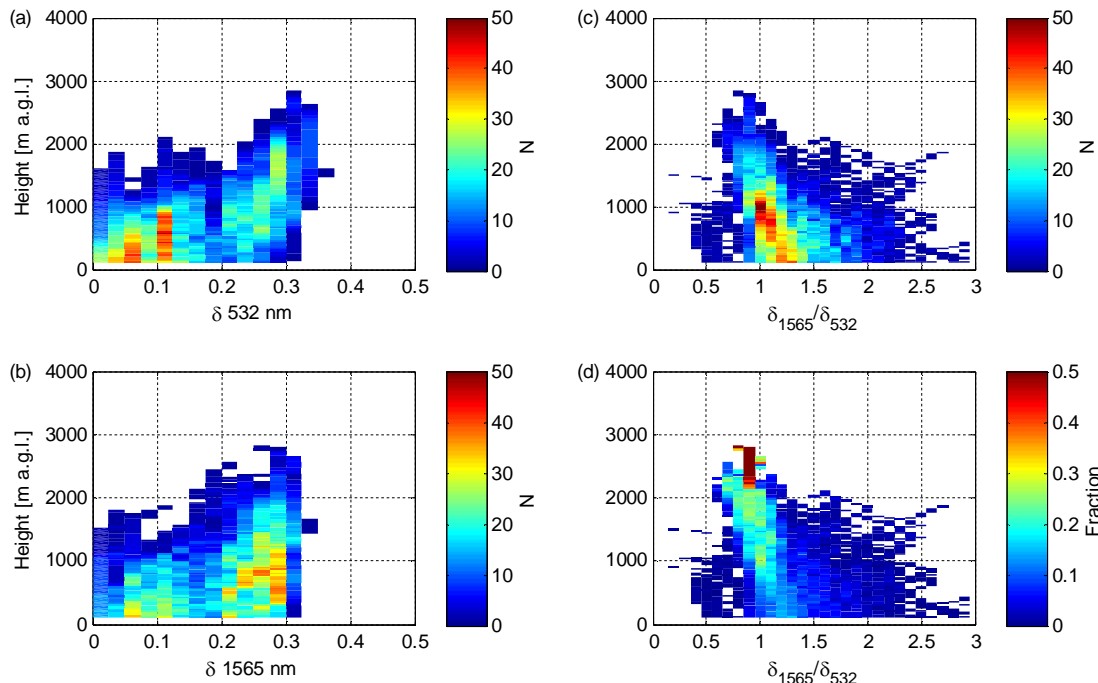


**Figure 16: 2D histograms of depolarization ratio and height at Limassol using 30 m vertical resolution smoothed by 300 m running mean. Only data for $\sigma_\delta < 0.01$ (at 1565 nm wavelength) is included. (a) Depolarization ratio at 532 nm. (b) Depolarization ratio at 1565 nm. (c) Ratio of depolarization ratios at 1565 nm and 532 nm. (d) Same as panel (c) but scaled with number of observations at each height.**
