# Peer review of "Aerosol particle depolarization ratio at 1565 nm measured with a Halo Doppler lidar"

_Atmospheric Chemistry and Physics, 2020_

## Referee Comment (RC1) · Anonymous Referee #1 · 12 Jan 2021

The authors present a new technique that enables depolarization measurements from a Halo Doppler lidar. The new product falls at 1565nm, an unexplored wavelength until now. Consequently, the scientific impact is rather high, while possible spread of the technique may increase the information from the Halo systems operating worldwide. The manuscript is also written in a clear way. My main criticism lies in the cloud calibration method that has been selected for the analysis as it poses many risks that are not properly addressed in the manuscript. I would recommend this manuscript for publication after some revisions.

—————General Comments—————

Lines 128-141: Cloud calibration has the following risks:

[Figure]

1) The co polar signal can be saturated, especially for low clouds. Then, the ratio in the cloud base is not reliable anymore. 2) Vertical filtering/smoothing usually generates artifacts near the cloud base, especially for sharp low clouds. Are the profiles that were used for calibration smoothed? If yes, what kind of filter has been applied 3) There is always the multiple scattering issue that is discussed in the manuscript. Reducing the FOV by e.g. reducing the field stop radius if possible can reduce multiple scattering effects. 4) The measurements are not simultaneous! For aerosol variability 7 seconds are not important but for clouds a lot can change. This could negatively affect the ratios. Some information is provided in lines 136-138. What time scales are applied in the high time resolution data.

How did the authors deal with these issues?

Concerning the calibration, as stated above, using the same detector for both co and cross signals has the benefit that the ratio of the co and cross attenuated backscatter profiles gives the volume linear depolarization ratio (VLDR) directly. However, it seems that the authors use the SNR instead of the signals. In that case, to my understanding, the ratio of the 2 SNR is no longer the VLDR. Why not use the attenuated backscatter ratio directly for the volume linear depolarization ratio calculation?

There is always the case that depolarizing effects are introduced by the receiver and by the laser (depending on the emission purity) and the emission optics, but these can be accounted for with the GHK formalism introduced by Freudenthaler et al 2016.

The polarizer bleed through can be calculated in the laboratory. In addition, such values are usually provided by the manufacturers. Then, this effect can be taken into account into the K parameter of the GHK formalism (see Freudenthaler et al. 2016).

Finally, is the molecular atmosphere observable with HALO? If yes, the VLDR ratio in the molecular region should agree with the theoretical molecular VLDR at 1565nm with the respect to the FWHM around the mainline that is collected by the detector (see Behrend et al. 2002). Is it possible to perform such a test? The GHK correction is then

applied to correct any offsets.

—————Comments—————

Lines 54: It has been recently shown (Gialitaki et al 2020) that soot aggregates can assume the near spherical shape. In their study they present the depolarization ratio values that are expected per wavelength.

Line 91: Strong H2O absorption takes place in the spectral region near 1565nm. Does this specific wavelength fall in a low absorption region? Are there any H2O vertical extinction corrections required?

Lines 95-96: A single channel has been used for the depolarization measurements. The benefit of this setup is that it does not require a calibration factor. However, is is difficult to achieve similar order of magnitude co and cross polarized signals. This can result to non linear amplification in the detection since the detector might not operate optimally for such a demanding dynamic range. In 2 channel setups a neutral density filter is used in front of the co polarized channel to bring its naturally higher intensity to the crossed polarized levels. Have the authors checked for effects in their setup (e.g. saturation and/or clipping in the co signal or increased noise due to a weak cross signal).

Lines 112-115: The authors should provide more information and references on how the SNR is calculated. Is it the raw signal divided by a constant noise level? Is the noise vertically resolved?

Line 115 'by averaging the SNR': Is the SNR averaged or the signal in order to reduce noise and increase the SNR?

Lines 117-120: A US standard atmosphere model is preferred here. This can introduce uncertainties in the retrievals The use of a dedicated radiosonde or a meteorological model is a much safer approach. Did the author compare the retrievals using the US standard atmosphere with a radiosonde?

Figure 2: A figure with the SNR (or signal) with the same vertical scale as the VLDR plot is missing here. The vertical scale of the left part of figure 2 makes it difficult to compare the two plots.

General figure comment: Please specify whether the depolarization ratio is the particle linear depolarization ratio or the volume linear depolarization ratio.

—————Recommendations—————

In order to optimize the VLDR retrievals a GHK correction can be applied. Optimally, the authors should measure the purity of their emission and also perform a Delta 90 calibration with a rotator or a rotating linear polarizer in front of their receiver box, if possible, to measure any diattenuation and/or retardation effects coming from the receiver. Then they can apply the measured properties to calculate the GHK parameters and then, obtain the corrected VLDR profile from the signal ratio.

Section 3.4 It would be interesting to take also into account the modeled wavelength dependence of such aerosols, at least for dust. The latest version of the OPAC database (Koepke et al. 2015) includes non spherical dust particles in three modes (ultra fine, coagulation, and coarse). The wavelength dependence is different depending on the size mode. This could be taken into account in the discussion here. Similar databases like MOPSMAP (Gasteiger et al. 2018) could be also taken into account.

—————References—————

– Behrendt, A. & Nakamura, T. Calculation of the calibration constant of polarization lidar and its dependency on atmospheric temperature Opt. Express, OSA, 2002, 10, 805-817

– Freudenthaler, V. About the effects of polarising optics on lidar signals and the $\Delta$ 90Âăcalibration Atmospheric Measurement Techniques, 2016, 9, 4181-4255

– Koepke, P.; Gasteiger, J. & Hess, M. Technical Note: Optical properties of desert aerosol with non-spherical mineral particles: data incorporated to OPAC Atmospheric

Chemistry and Physics, 2015, 15, 5947-5956

– Gasteiger, J. & Wiegner, M. MOPSMAP v1.0: a versatile tool for the modeling of aerosol optical properties Geoscientific Model Development, 2018, 11, 2739-2762

– Gialitaki, A.; Tsekeri, A.; Amiridis, V.; Ceolato, R.; Paulien, L.; Kampouri, A.; Gkikas, A.; Solomos, S.; Marinou, E.; Haarig, M.; Baars, H.; Ansmann, A.; Lapyonok, T.; Lopatin, A.; Dubovik, O.; Groß, S.; Wirth, M.; Tsichla, M.; Tsikoudi, I. & Balis, D. Is the near-spherical shape the "new black" for smoke? Atmospheric Chemistry and Physics, 2020, 20, 14005-14021

---

## Referee Comment (RC2) · Anonymous Referee #2 · 16 Jan 2021

In their manuscript the authors describe a new method to derive the particle linear depolarizatio ratio for HALO photonics lidar systems. The particle linear depolarization ratio is a very important property to distighuish different aerosol types. Thus, an additional method to derive this important property is of high significance. The manuscript is very well written and easy to follow. The technique and the results are very well presented. I suggest publication after some minor revisions.

Comments:

The authors present a new method to derive the particle linear depolarization ratio at a quite long lidar wavelength. Can the new method / the particle linear depolarization derived from the 1565 nm measurements be used stand alone for aerosol typing, or is its main purpose an extension of existing classification schemes to provide additional

information for a more robust classification.

Can the authors give a few more words on the calibration of the two signals / on the uncertainties resulting from their kind of calibration? The mean values of the two systems do not show a significant difference; does it represent a universial characteristic of the HALO Photonics systems? How often should the calibration be performed?

The authors show an important comparison of their results compared to former measurements. Especially with regard to the longer wavelength, a comparison with results from optical modelling would be interesting and is missing in this manuscript.

---

## Author Comment (AC1) · 26 Feb 2021

We would like to thank the referee for the constructive comments. Below are referee comments in **bold**, followed by our reply.

**The authors present a new technique that enables depolarization measurements from a Halo Doppler lidar. The new product falls at 1565nm, an unexplored wavelength until now. Consequently, the scientific impact is rather high, while possible spread of the technique may increase the information from the Halo systems operating worldwide. The manuscript is also written in a clear way. My main criticism lies in the cloud calibration method that has been selected for the analysis as it poses many risks that are not properly addressed in the manuscript. I would recommend this manuscript for publication after some revisions.**

**—————General Comments—————**

**Lines 128-141: Cloud calibration has the following risks:**

**1) The co polar signal can be saturated, especially for low clouds. Then, the ratio in the cloud base is not reliable anymore. 2) Vertical filtering/smoothing usually generates artifacts near the cloud base, especially for sharp low clouds. Are the profiles that were used for calibration smoothed? If yes, what kind of filter has been applied 3) There is always the multiple scattering issue that is discussed in the manuscript. Reducing the FOV by e.g. reducing the field stop radius if possible can reduce multiple scattering effects. 4) The measurements are not simultaneous! For aerosol variability 7 seconds are not important but for clouds a lot can change. This could negatively affect the ratios. Some information is provided in lines 136-138. What time scales are applied in the high time resolution data.**
**How did the authors deal with these issues?**

We agree that cloud calibration has risks, and we have expanded the discussion in the manuscript based on the referee's comments. To answer the comments above:

1) Indeed, co polar signal can get saturated. We have observed this for XR devices, but not for the less powerful Stream Line versions used here. Saturation can be easily seen as non-linearity in a scatterplot of co and cross polar SNR, if co-polar SNR covers a wide range of signal strengths. It could also be seen as broadening of the distribution in Fig. 3. For the devices we have investigated, standard deviation less than 0.01 for Fig. 3 distribution is typical when no saturation occurs.
2) As signal from cloud base is strong, no vertical smoothing is needed. We use the original 30 m range resolution.
3) Unfortunately, we cannot change the optical path of the instrument. However, the instrument telescope design has a matched FOV and divergence of 33 µrad and consequently the multiple scattering effects are relatively small as seen in Fig. 2b: at cloud base the ratio of cross to co SNR is 0.0102 and at the next gate inside cloud 0.0116. Only at the third in-cloud range gate the ratio of cross to co SNR increases to 0.0224. The increase by 0.0014 in 30 m is small compared e.g. to the system considered by Donovan et al. (2015), where in-cloud multiple scattering increases $\delta$ from 0 to 0.05 in approx. 50 m.
4) Yes, the integration time is a compromise between aerosol and cloud measurements we have to make. The same time resolution is maintained through the campaign. We try to minimise the effects of relatively poor time resolution by choosing cases, where cloud base remains at the same altitude (within lidar resolution) for some tens of minutes. Non-simultaneous

measurements broaden the distribution in Fig. 3. This uncertainty is propagated into the final uncertainty in the depolarization ratio as indicated in Equations 2 and 3.

We have modified lines 130-141 as:

"Spherical cloud droplets do not polarize the directly back-scattered radiation (e.g. Liou and Schotland, 1971) and thus non-zero $\delta^*$ at liquid cloud base is an indication of incomplete extinction (or bleed-through) in the lidar internal polarizer. However, measurement of $\delta^*$ at cloud base can be biased by signal saturation or changes in cloud properties between co- and cross-polar measurement. Furthermore, multiple scattering results in increasing depolarization signal inside a liquid cloud (e.g. Liou and Schotland, 1971). This increase in in-cloud $\delta^*$ is clearly seen in Fig. 2c: at cloud base $\delta^*$ is 0.0102 and at the next gate 30 m deeper inside the cloud $\delta^*$ has increased to 0.0116.

The magnitude of the multiple scattering effect on depolarization ratio depends on both cloud and lidar properties (e.g. Donovan et al., 2015). In Halo Stream Line lidars the instrument telescope design has a matched FOV and divergence of 33 µrad (Table 1) and consequently the effect is small: in Fig. 2c $\delta^*$ increases by 0.0014 in 30 m. For instance, for the system modelled by Donovan et al. (2015) in-cloud multiple scattering increases depolarization ratio from 0 to 0.05 in approx. 50 m. Nevertheless, to minimize the effect of multiple scattering we only consider $\delta^*$ at the cloud base for the determination of the average bleed-through and use measurements in several clouds.

For low-level clouds, we have observed saturation of the co-polar signal in the more powerful Stream Line XR instruments. Signal saturation at liquid cloud base is readily identified as non-linear relationship between co- and cross-polar SNR. For the measurement cases analysed here, we did not observe indications of saturation. Furthermore, we note that $\delta^*$ at cloud-base should be determined with the highest possible time resolution to ensure that both co- and cross-polar measurements represent the same part of the cloud. In practice, integration time is kept constant during a measurement campaign, and was set as a compromise between temporal resolution and signal strength. We mitigate the effect of poor time resolution by choosing cases, where cloud base remains at the same altitude (within lidar resolution) for some tens of minutes and thus one can assume temporal homogeneity. No vertical smoothing is applied in calculating $\delta^*$, as signal at cloud base is strong compared to aerosol signal. Finally, it should be noted that, especially in higher latitudes, it is not always trivial to find purely liquid phase clouds. Typically, mixed-phase clouds can be distinguished by the depolarizing features of ice crystals. I.e., in the histogram of $\delta^*$ at cloud-base a secondary peak with higher $\delta^*$ than liquid clouds would occur, which was not the case for our study."

**Concerning the calibration, as stated above, using the same detector for both co and cross signals has the benefit that the ratio of the co and cross attenuated backscatter profiles gives the volume linear depolarization ratio (VLDR) directly. However, it seems that the authors use the SNR instead of the signals. In that case, to my understanding, the ratio of the 2 SNR is no longer the VLDR. Why not use the attenuated backscatter ratio directly for the volume linear depolarization ratio calculation?**

For Halo, the primary parameters that are provided by the firmware are radial velocity and SNR, which is processed by the firmware into attenuated backscatter as the third output parameter.

For a coherent Doppler lidar attenuated backscatter is obtained as

$$\beta(z) = A\frac{\text{SNR}(z)}{T_f(z)} \tag{1},$$

where z is range from instrument, A incorporates system-specific constants (e.g. beam energy and receiver bandwidth) and $T_f(z)$ is telescope focus function, which includes range correction (Frehlich and Kavaya,1991; Pentikäinen et al., 2020). Now, $T_f(z)$ is equal for both polarities and thus the ratio of SNR is equal to the ratio of attenuated backscatter. Using the SNR ratio instead of attenuated backscatter we do not need to calculate attenuated backscatter from the cross-polar SNR and the processing chain can be kept a little simpler.

Unfortunately, SNR from the firmware often has a small bias (Manninen et al., 2016). To correct for this bias, we use the SNR post-processing method by Vakkari et al. (2019). The post-processing ensures that both co- and cross-polar SNR have well-defined noise level, i.e. SNR is zero if there is no aerosol or cloud signal. This is readily verified with the fully scanning system with measurements pointing down.

We have modified lines 112-115 to clarify this:

"Halo Stream Line lidars provide three parameters along the beam direction: radial Doppler velocity, signal-to-noise ratio (SNR), and attenuated backscatter (β), which is calculated from SNR taking into account the telescope focus. For a coherent Doppler lidar attenuated backscatter is obtained as

$$\beta(z) = A\frac{SNR(z)}{T_f(z)}, \tag{1}$$

where z is range from instrument, A incorporates system-specific constants and $T_f(z)$ is telescope focus function, which includes range correction (Frehlich and Kavaya,1991; Pentikäinen et al., 2020).

A background check to determine range-resolved background noise level is performed automatically once per hour. The raw signal from atmospheric measurement is then divided by this noise level in the firmware and returned as SNR (see Vakkari et al., 2019). We post-processed SNR according to Vakkari et al. (2019), which ensures that both co- and cross-polar SNR have an unbiased noise level, i.e. SNR is 0 when there is no signal (c.f. Manninen et al., 2016). Furthermore, the post-processing is essential to be able to further reduce the instrumental noise floor by averaging the SNR (Vakkari et al., 2019). After post-processing SNR, β is calculated with Eq. 1."

**There is always the case that depolarizing effects are introduced by the receiver and by the laser (depending on the emission purity) and the emission optics, but these can be accounted for with the GHK formalism introduced by Freudenthaler et al 2016. The polarizer bleed through can be calculated in the laboratory. In addition, such values are usually provided by the manufacturers. Then, this effect can be taken into account into the K parameter of the GHK formalism (see Freudenthaler et al. 2016).**

We agree that the GHK formalism (Freudenthaler et al., 2016) is an excellent tool for calibrating depolarization ratio and we use it for the PollyXT measurements. However, GHK formalism requires access to the optical path of the instrument, which we do not have for the Halo instruments. For instance, we cannot extract the polarizer for laboratory tests or add a calibrator in the optical path. Furthermore, the information from the lidar manufacturer on the polarizer performance is not quantitative, merely an acknowledgement that it is not perfect. Therefore, we obtain the depolarization ratio from the Halo with the highest quality possible, but knowing that it is not yet mature for meeting the ACTRIS standard for depolarization measurements. Nevertheless, this was

not the focus of the manuscript as here we present a first assessment of the depolarization capabilities of such Doppler lidar system far from being used as quality-assured standard ACTRIS output.

We have clarified the limitations of Halo construction on lines 128-129:

"The construction of Halo Stream Line lidars does not include a calibrator for depolarization channel, unlike PollyXT lidars for instance (Engelmann et al., 2016). Furthermore, the user cannot change the optical path to include a calibrator or check the depolarizing effects of the individual components. Therefore, we are limited to evaluating the Halo depolarization ratio at liquid cloud base."

**Finally, is the molecular atmosphere observable with HALO? If yes, the VLDR ratio in the molecular region should agree with the theoretical molecular VLDR at 1565nm with the respect to the FWHM around the mainline that is collected by the detector (see Behrend et al. 2002). Is it possible to perform such a test? The GHK correction is then applied to correct any offsets.**

Molecular return cannot be observed with Halo, as stated on lines 120-122. Operating at low pulse energy, even aerosol layers observable with PollyXT are often too weak for Halo. This is seen e.g. in Fig. 5a, where attenuated backscatter from Halo is not available above 5.7 km, although PollyXT indicates aerosol layers up to 7 km.

—————Comments—————

**Lines 54: It has been recently shown (Gialitaki et al 2020) that soot aggregates can assume the near spherical shape. In their study they present the depolarization ratio values that are expected per wavelength.**

Thank you for pointing out this reference. We have added a sentence on line 55:

"Recently, Gialitaki et al. (2020) modelled smoke as near-spherical submicron particles and found good agreement with the observed spectral dependency of depolarization ratio."

**Line 91: Strong H2O absorption takes place in the spectral region near 1565nm. Does this specific wavelength fall in a low absorption region? Are there any H2O vertical extinction corrections required?**

According to HITRAN (https://hitran.iao.ru/, last access 17 February 2021) transmittance at 1565 nm is more than 0.9999. I.e. there is no absorption by H2O at this wavelength.

**Lines 95-96: A single channel has been used for the depolarization measurements. The benefit of this setup is that it does not require a calibration factor. However, is is difficult to achieve similar order of magnitude co and cross polarized signals. This can result to non linear amplification in the detection since the detector might not operate optimally for such a demanding dynamic range. In 2 channel setups a neutral density filter is used in front of the co polarized channel to bring its naturally higher intensity to the crossed polarized levels. Have the authors checked for effects in their setup (e.g. saturation and/or clipping in the co signal or increased noise due to a weak cross signal).**

We have observed saturation in the co signal for cloud base, but only in the more powerful Stream Line XR instruments, which is now mentioned on lines 130-141. For aerosols, we have not observed signal saturation and do not expect it to occur, given that saturation is rare even for the much stronger cloud return.

For low depolarization ratio aerosol, the noise level in the cross signal is substantial, but can be reduced by longer averaging time to some extent. In any case, the cross signal noise level is taken into account when calculating the uncertainty in the depolarization ratio.

**Lines 112-115: The authors should provide more information and references on how the SNR is calculated. Is it the raw signal divided by a constant noise level? Is the noise vertically resolved?**

SNR is provided by the firmware, which is not open software. Our best understanding of the SNR calculation in the firmware is presented in Vakkari et al. (2019). In short, the raw signal is divided by range-resolved noise level, which is obtained from the previous (hourly) background check by the firmware. Additionally, both raw signals (atmospheric return and background check) seem to be scaled by scalars in the firmware (Vakkari et al., 2019).

The post-processing improves the accuracy of the noise level from the background check and corrects any bias in the scaling factors. We have added this information on lines 113-114:

"A background check to determine range-resolved background noise level is performed automatically once per hour. The raw signal from atmospheric measurement is then divided by this noise level in the firmware and returned as SNR (see Vakkari et al., 2019)."

**Line 115 'by averaging the SNR': Is the SNR averaged or the signal in order to reduce noise and increase the SNR?**

SNR is averaged as we do not have raw signal, which would not be scaled by the noise level. We hope that the modifications to lines 112-115 make this clearer (please see our response to the second general comment above).

**Lines 117-120: A US standard atmosphere model is preferred here. This can introduce uncertainties in the retrievals The use of a dedicated radiosonde or a meteorological model is a much safer approach. Did the author compare the retrievals using the US standard atmosphere with a radiosonde?**

Here, we have used the US standard atmosphere to estimate the molecular contribution to the signal to show that it cannot be observed. We agree that using radiosondes or numerical weather prediction (NWP) models to estimate the profile of number density is more accurate, particularly at shorter wavelengths, and we have also checked the values at 1565 nm in the cases presented here using NWP profiles. However, in the text, we think it is clearer to provide a single value, noting that it is from a representative atmosphere.

We have reformatted lines 117-122 to make this more apparent:

"Given the long wavelength and low pulse energy, no contribution from molecular scattering is observed in the signal. At 1565 nm the molecular backscatter coefficient is about $1.9 \times 10^{-8}$ m$^{-1}$ sr$^{-1}$

at mean sea level, using mean values for the atmospheric number density taken from the U.S. Standard Atmosphere, 1976 (COESA 1976). The two-way atmospheric transmittance at 1565 nm is still 0.9994 at 2 km altitude above a lidar situated at mean sea level. Hence, the measured depolarization ratio can be safely assumed to represent the particle linear depolarization ratio."

**Figure 2: A figure with the SNR (or signal) with the same vertical scale as the VLDR plot is missing here. The vertical scale of the left part of figure 2 makes it difficult to compare the two plots.**

We have added a third panel in Fig. 2 with the same vertical scale as depolarization ratio.

[Figure]

**Figure 1: Profiles at Limassol, Cyprus, on 2 May 2017 at 12:08 UTC. (a) Co- and cross-polar SNR. A liquid cloud at approx. 800 m a.g.l. results in full attenuation of signal. Below cloud layer aerosol signal is visible. Above 1 km variability in SNR is due to instrumental noise only. (b) The same as panel (a), but limited to lowest 1 km a.g.l.. (c) Ratio of cross-polar SNR to co-polar SNR up to 1 km a.g.l. calculated from profiles in panel (a). Error bars represent uncertainty due to instrumental noise estimated from SNR at > 1 km a.g.l. in panel (a).**

**General figure comment: Please specify whether the depolarization ratio is the particle linear depolarization ratio or the volume linear depolarization ratio.**

We have updated figure captions with this information. We have also defined the raw ratio of Halo cross-SNR to co-SNR as δ* in Figures 2 and 3 and in the text to avoid confusion with the Halo depolarization ratio after bleed-through correction.

—————Recommendations—————

**In order to optimize the VLDR retrievals a GHK correction can be applied. Optimally, the authors should measure the purity of their emission and also perform a Delta 90 calibration with a rotator or a rotating linear polarizer in front of their receiver box, if possible, to measure any diattenuation and/or retardation effects coming from the receiver. Then they can apply the measured properties to calculate the GHK parameters and then, obtain the corrected VLDR profile from the signal ratio.**

We agree that it would be best to perform the Δ90°-calibration. Unfortunately, with a system that uses a single lens for sending and receiving pulses (see lines 106-107 in the manuscript) we cannot place a polarizer in front of the receiver, and we are left with the cloud base measurements as the only option for time being.

**Section 3.4 It would be interesting to take also into account the modeled wavelength dependence of such aerosols, at least for dust. The latest version of the OPAC database (Koepke et al. 2015) includes non spherical dust particles in three modes (ultra fine, coagulation, and coarse). The wavelength dependence is different depending on the size mode. This could be taken into account in the discussion here. Similar databases like MOPSMAP (Gasteiger et al. 2018) could be also taken into account.**

We have added a new figure in Section 4 including spectral dependency of depolarization ratio modelled by MOPSMAP (Gasteiger and Wiegner, 2018) for desert dust aerosol and an Aeronet inversion up to 1640 nm by Toledano et al. (2019). Here, we would like to keep the manuscript focused on measurements and decided to leave more detailed model comparison for future studies.

The new figure and related discussion added on line 309 are:

"Spectral dependency of depolarization ratio modelled with MOPSMAP (Gasteiger and Wiegner, 2018) for desert dust aerosol from OPAC database (Koepke et al., 2015) agrees reasonably well with the Saharan dust case on 21 April 2017 in this study (Fig. 17). On the other hand, the sun photometer based retrieval by Toledano et al. (2019) for long-range transported Saharan dust over Barbados indicates a little lower depolarization ratio of 0.19 at 1640 nm compared to this study at 1565 nm (Fig. 17). The lower depolarization ratio at 1640 nm over Barbados is reasonable considering the much longer transport compared to this study."

[Figure]

**Figure 2: Particle linear depolarization ratio as function of wavelength for dust observations in Table 2. Additionally, spectral dependency modelled with MOPSMAP based on OPAC database for desert dust (Koepke et al., 2015; Gasteiger and Wiegner, 2018) and Aeronet inversion by Toledano et al. (2019) are included.**

—————References—————

– Behrendt, A. & Nakamura, T. Calculation of the calibration constant of polarization

lidar and its dependency on atmospheric temperature Opt. Express, OSA, 2002, 10, 805-817

– Freudenthaler, V. About the effects of polarising optics on lidar signals and the 90Å˘ acalibration Atmospheric Measurement Techniques, 2016, 9, 4181-4255

– Koepke, P.; Gasteiger, J. & Hess, M. Technical Note: Optical properties of desert aerosol with non-spherical mineral particles: data incorporated to OPAC Atmospheric Chemistry and Physics, 2015, 15, 5947-5956

– Gasteiger, J. & Wiegner, M. MOPSMAP v1.0: a versatile tool for the modeling of aerosol optical properties Geoscientific Model Development, 2018, 11, 2739-2762

– Gialitaki, A.; Tsekeri, A.; Amiridis, V.; Ceolato, R.; Paulien, L.; Kampouri, A.; Gkikas, A.; Solomos, S.; Marinou, E.; Haarig, M.; Baars, H.; Ansmann, A.; Lapyonok, T.; Lopatin, A.; Dubovik, O.; Groß, S.; Wirth, M.; Tsichla, M.; Tsikoudi, I. & Balis, D. Is the near-spherical shape the "new black" for smoke? Atmospheric Chemistry and Physics, 2020, 20, 14005-14021

Frehlich, R. G. and Kavaya, M. J.: Coherent laser radar performance for general atmospheric refractive turbulence, Appl. Opt., 30(36), 5325–5352, doi:10.1364/AO.30.005325, 1991.

Manninen, A. J., O'Connor, E. J., Vakkari, V. and Petäjä, T.: A generalised background correction algorithm for a Halo Doppler lidar and its application to data from Finland, Atmos. Meas. Tech., 9(2), 817–827, https://doi.org/10.5194/amt-9-817-2016, 2016.

Pentikäinen, P., O'Connor, E. J., Manninen, A. J. and Ortiz-Amezcua, P.: Methodology for deriving the telescope focus function and its uncertainty for a heterodyne pulsed Doppler lidar, Atmos. Meas. Tech., 13(5), 2849–2863, doi:10.5194/amt-13-2849-2020, 2020.

Toledano, C., Torres, B., Velasco-Merino, C., Althausen, D., Groß, S., Wiegner, M., Weinzierl, B., Gasteiger, J., Ansmann, A., González, R., Mateos, D., Farrel, D., Müller, T., Haarig, M. and Cachorro, V. E.: Sun photometer retrievals of Saharan dust properties over Barbados during SALTRACE, Atmos. Chem. Phys., 19(23), 14571–14583, doi:10.5194/acp-19-14571-2019, 2019.

Vakkari, V., Manninen, A. J., O'Connor, E. J., Schween, J. H., van Zyl, P. G. and Marinou, E.: A novel post-processing algorithm for Halo Doppler lidars, Atmos. Meas. Tech., 12(2), 839–852, doi:10.5194/amt-12-839-2019, 2019.

---

## Author Comment (AC2) · 26 Feb 2021

We would like to thank the referee for the constructive comments. Below are referee comments in **bold**, followed by our reply.

**In their manuscript the authors describe a new method to derive the particle linear depolarizatio ratio for HALO photonics lidar systems. The particle linear depolarization ratio is a very important property to distighuish different aerosol types. Thus, an additional method to derive this important property is of high significance. The manuscript is very well written and easy to follow. The technique and the results are very well presented. I suggest publication after some minor revisions.**

**Comments:**

**The authors present a new method to derive the particle linear depolarization ratio at a quite long lidar wavelength. Can the new method / the particle linear depolarization derived from the 1565 nm measurements be used stand alone for aerosol typing, or is its main purpose an extension of existing classification schemes to provide additional information for a more robust classification.**

Most aerosol typing algorithms are based on a combination of depolarization ratio and other parameters such as lidar ratio or Ångström exponent, as e.g. marine and polluted air masses can have very similar depolarization ratio (e.g. Illingworth et al., 2015; Baars et al., 2016). Thus, we see the new wavelength mostly as an extension of the current suite of measurements. However, if there is prior knowledge of prevailing aerosols (e.g. volcanic ash transport) also stand-alone measurements can provide useful aerosol typing.

We have added following paragraph in the conclusions on line 334:

"For aerosol typing, adding particle linear depolarization ratio at 1565 nm to shorter wavelengths can help to distinguish biomass burning aerosols from dust, as much stronger spectral dependency has been observed for elevated biomass burning aerosols than for dust (e.g. Haarig et al., 2017, 2018; Hu et al., 2019). In case there is prior knowledge of prevailing aerosols, such as transport of volcanic ash, even stand-alone particle linear depolarization ratio measurements with Halo Doppler lidars can probably provide useful information for aerosol typing."

**Can the authors give a few more words on the calibration of the two signals / on the uncertainties resulting from their kind of calibration? The mean values of the two systems do not show a significant difference; does it represent a universial characteristic of the HALO Photonics systems? How often should the calibration be performed?**

We have added a more detailed discussion on the calibration in response to Reviewer 1. In our experience from seven different Halo Photonics systems in Finland the bleed-through is typically less than 0.02. However, one of the XR systems has considerably higher bleed-through, though that can be partially attributed to higher uncertainty in the background noise level for XR systems (see Vakkari et al., 2019).

As the optical path is made of fibre-optic components, we do not expect significant temporal variation in the bleed-through. Continuous measurements in Finland have also shown that the bleedthrough remains constant for several years at least. However, we recommend to check the bleed-through monthly or after an instrument is moved to be sure.

We have added following on line 146:

"The mean cloud base δ* observed for these two systems in Fig. 3 are well in line with our experience with these and five other Stream Line and Stream Line XR systems in Finland, where cloud base δ* typically ranges from 0.01 to 0.02."

And on line 149:

"This is also our experience with Halo systems in Finland since 2016, but we recommend to check the bleed-through monthly or after an instrument is moved to a new location."

**The authors show an important comparison of their results compared to former measurements. Especially with regard to the longer wavelength, a comparison with results from optical modelling would be interesting and is missing in this manuscript.**

We have added a new figure in Section 4 including spectral dependency of depolarization ratio modelled by MOPSMAP (Gasteiger and Wiegner, 2018) for desert dust aerosol and an Aeronet inversion up to 1640 nm by Toledano et al. (2019). Here, we would like to keep the manuscript focused on measurements and decided to leave more detailed model comparison for future studies.

The new figure and related discussion added on line 309 are:

[revised manuscript text omitted]